# Fast two-photon imaging of subcellular voltage dynamics in neuronal tissue with genetically encoded indicators

Simon Chamberland[1†], Helen H Yang[2†], Michael M Pan[3,4], Stephen W Evans[2,3,4], Sihui Guan[5], Mariya Chavarha[2,3,4], Ying Yang[2,3,4], Charleen Salesse[1], Haodi Wu[6], Joseph C Wu[6], Thomas R Clandinin[2*], Katalin Toth[1*], Michael Z Lin[2,3,4], François St-Pierre[3,4,5*‡]

[1]Department of Psychiatry and Neuroscience, Quebec Mental Health Institute, Université Laval, Québec, Canada; [2]Department of Neurobiology, Stanford University, Stanford, United States; [3]Department of Bioengineering, Stanford University, Stanford, United States; [4]Department of Pediatrics, Stanford University, Stanford, United States; [5]Department of Neuroscience, Baylor College of Medicine, Houston, United States; [6]Stanford Cardiovascular Institute, Stanford University, Stanford, United States

*For correspondence: trc@ stanford.edu (TRC); katalin.toth@ fmed.ulaval.ca (KT); francois.st-pierre@bcm.edu (FS-P)

[†]These authors contributed equally to this work

Present address: [‡]Department of Neuroscience, Baylor College of Medicine, Houston, United States

**Abstract** Monitoring voltage dynamics in defined neurons deep in the brain is critical for unraveling the function of neuronal circuits but is challenging due to the limited performance of existing tools. In particular, while genetically encoded voltage indicators have shown promise for optical detection of voltage transients, many indicators exhibit low sensitivity when imaged under two-photon illumination. Previous studies thus fell short of visualizing voltage dynamics in individual neurons in single trials. Here, we report ASAP2s, a novel voltage indicator with improved sensitivity. By imaging ASAP2s using random-access multi-photon microscopy, we demonstrate robust single-trial detection of action potentials in organotypic slice cultures. We also show that ASAP2s enables two-photon imaging of graded potentials in organotypic slice cultures and in *Drosophila*. These results demonstrate that the combination of ASAP2s and fast two-photon imaging methods enables detection of neural electrical activity with subcellular spatial resolution and millisecond-timescale precision.

## Introduction

Neurons represent, process, and propagate information by controlling the potential across their plasma membrane. Methods to measure electrical activity are thus central to efforts to understand computations in the brain. While electrophysiological approaches have been used for decades, the need to track genetically defined neuronal subpopulations has motivated the development and application of protein-based fluorescent reporters of neural activity. In contrast to electrode-based methods, these optophysiological indicators also allow monitoring without placement of physical probes near or in the neurons of interest. Thus, they can enable easier and less invasive measurement of activity from individual neurons and from their subcellular compartments such as axons and dendrites.

Genetically encoded calcium indicators are commonly used to detect the forms of neuronal activity that trigger calcium flux into neurons (*Grienberger and Konnerth, 2012*; *Tian et al., 2012*; *Lin and Schnitzer, 2016*). For example, recent calcium indicators such as GCaMP6f can detect single action potentials (APs) in cell bodies and synaptic responses in spines (*Chen et al., 2013*). However,

calcium concentration is not directly related to membrane potential, and therefore cannot be used to follow voltage changes that do not result in substantial calcium fluxes. Calcium indicators thus cannot effectively report hyperpolarizations and somatic subthreshold depolarizations in many neuronal cell types. The slow kinetics of calcium indicators and of calcium transients also limit the ability of calcium indicators to report voltage changes with high temporal precision and to track fast trains of action potentials (*Koester and Sakmann, 2000*,*Helmchen et al., 1996*; *Theis et al., 2016*).

To follow membrane potential dynamics more accurately, fluorescent indicators that monitor transmembrane voltage rather than calcium concentration have been developed (*Lin and Schnitzer, 2016*; *Knöpfel et al., 2015*; *Yang and St-Pierre, 2016*). These genetically encoded voltage indicators (GEVIs) have been used to image activity of neuronal populations in vivo (*Akemann et al., 2010*, *2012*; *Scott et al., 2014*; *Carandini et al., 2015*; *Mutoh et al., 2015*; *Shimaoka et al., 2017*) and of single neurons within intact brain tissue (*Ahrens et al., 2012*; *Akemann et al., 2013*; *Flytzanis et al., 2014*; *Gong et al., 2014*, *2015*; *Storace et al., 2015*) using widefield one-photon illumination. However, two-photon microscopy is the preferred modality for in vivo studies as it allows imaging with deeper tissue penetration and with lower background fluorescence and photo-toxicity outside of the focal point (*Svoboda and Yasuda, 2006*; *Prevedel et al., 2016*). GEVI-based voltage imaging in intact tissue with two-photon excitation has only been demonstrated for monitoring neural activity with either low temporal resolution (less than 100 Hz), low spatial resolution (averaging over ensembles of cells), or both (*Ahrens et al., 2012*; *Akemann et al., 2013*; *Storace et al., 2015*; *Yang et al., 2016*). Robust single-trial two-photon GEVI imaging with millisecond-timescale and cellular or subcellular resolution in intact neuronal tissue has not yet been reported.

Here, we demonstrate GEVI-based two-photon voltage imaging with millisecond-timescale precision, subcellular resolution, and the ability to simultaneously monitor spatially segregated locations. We first describe the development and characterization of ASAP2s, a higher-sensitivity variant of the GFP-based indicator ASAP1 (*St-Pierre et al., 2014*). We next compare ASAP2s with other GEVIs for two-photon imaging of light-evoked subcellular voltage dynamics in living flies. Having evaluated ASAP2s in vitro and in vivo, we deploy this new indicator to image voltage dynamics using random-access multi-photon microscopy in organotypic hippocampal slice cultures, where we demonstrate the ability of ASAP2s to detect action potentials, subthreshold depolarizations, and hyperpolarizations in individual cells in single trials. Finally, we show that ASAP2s can report action potential back-propagation in dendritic arbors and that this new GEVI enables single-trial, single-voxel spike detection at the soma.

## Results

### Development and characterization of ASAP2s

We aimed to improve and characterize the performance of GEVIs for reporting rapid neuronal electrical activity using two-photon microscopy in brain tissue. A previous study demonstrated that the two-photon properties of GEVIs can differ from their one-photon characteristics, with several indicators exhibiting response amplitudes that were more than 75% smaller with two-photon than with one-photon excitation (*Brinks et al., 2015*). In contrast, the ASAP1 indicator (*St-Pierre et al., 2014*) was unique in maintaining similar response amplitudes under one- and two-photon illumination. We therefore considered ASAP1 as a promising template for two-photon imaging.

The probability of correctly identifying spikes from optical noise is higher for sensors with larger response amplitudes, greater brightness, and longer signal decay (slower off-rate) (*Wilt et al., 2013*). We therefore sought to develop sensors with improvements in one or more of these characteristics. We performed rational mutagenesis in the transmembrane four-helix voltage-sensing domain (VSD) of ASAP1 (*Figure 1A*), focusing on residues important for sensing or responding to changes in the electric field. We tested the resulting mutants by patch-clamp electrophysiology in HEK293A cells. We first mutated a conserved site in the first helix (S1) that is thought to be part of a hydrophobic plug that focuses the electric field in homologous proteins (*Lacroix and Bezanilla, 2012*). However, none of the mutants exhibited improved response amplitudes to rapid, millisecond-timescale changes in electrical activity (*Figure 1—figure supplement 1A–H*). We then targeted positively charged residues in the fourth helix (S4) responsible for sensing the transmembrane electrical field (*Bezanilla, 2008*). ASAP1 with an R415Q mutation, which neutralizes one of these sensing

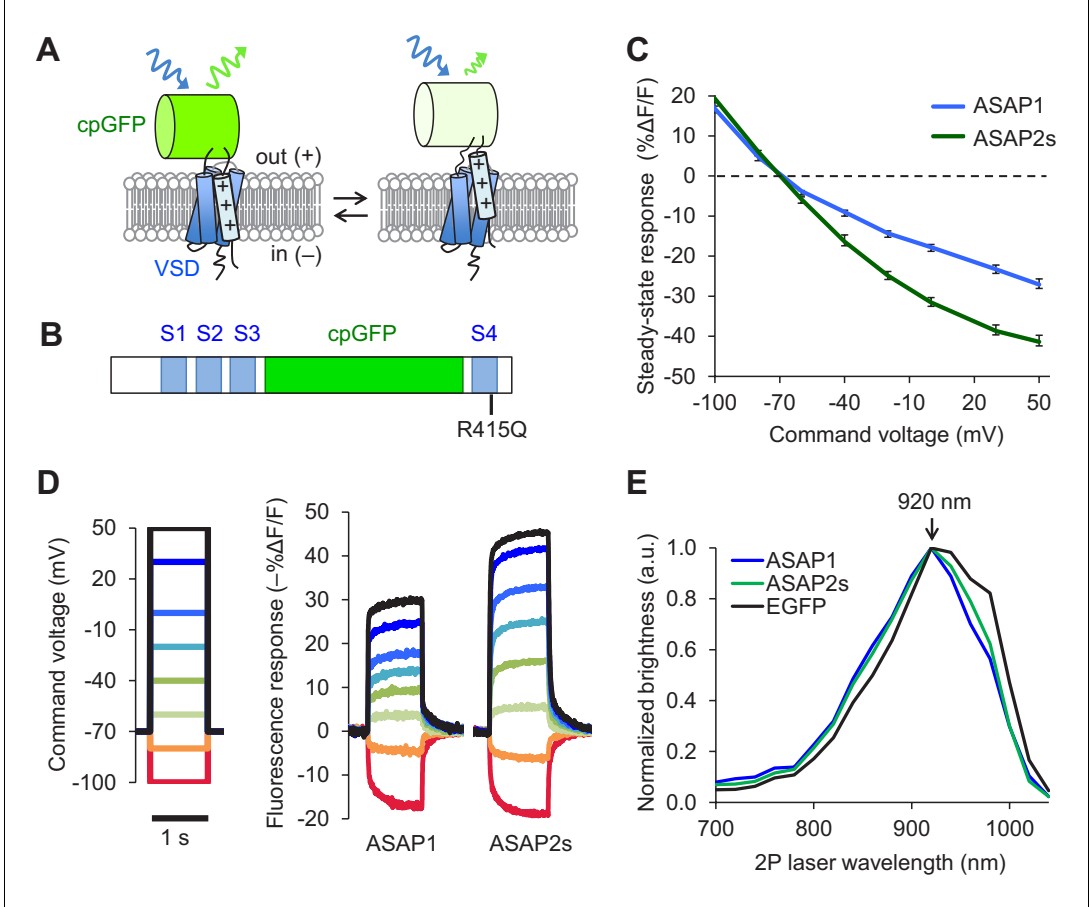

**Figure 1.** Design and in vitro characterization of ASAP2s. (**A**) In ASAP-type sensors, voltage-induced movement of a positively charged transmembrane helix of a voltage-sensing domain (VSD) is thought to perturb the protonation state of a circularly permuted GFP (cpGFP), resulting in changes in fluorescence emission. (**B**) Schematic diagram of ASAP1, showing the VSD transmembrane domains (S1 to S4, blue), cpGFP, and the location of the new mutation in ASAP2s (R415Q). (**C**) Mean fluorescence responses to voltage steps (n = 6 HEK293A cells for ASAP1 and n = 5 for ASAP2s). Error bars are standard error of the mean (SEM). Responses are reported as the fluorescence change ($\Delta$F) normalized by the initial fluorescence (F), expressed as a percentage of the initial fluorescence (% $\Delta$F/F). (**D**) Representative fluorescence responses of ASAP1 and ASAP2s to voltage steps from −100 to 50 mV. Responses were measured at 5 ms intervals and were normalized to the fluorescence at the −70 mV holding potential. (**E**) Two-photon excitation spectra of ASAP1, ASAP2s, and EGFP. All proteins were expressed in HEK293-Kir2.1 cells with resting membrane potential of ~-77 mV. Brightness was evaluated every 20 nm from 700 to 1040 nm. Each spectrum was normalized to its peak brightness. Traces are the mean of >30 cells.

The following figure supplements are available for figure 1:

**Figure supplement 1.** In vitro characterization of candidate GEVIs based on ASAP1.

**Figure supplement 2.** Brightness of ASAP sensors in immortalized cells.

**Figure supplement 3.** Photostability of ASAP sensors in immortalized cells under one-photon illumination.

**Figure supplement 4.** Photostability of ASAP sensors in immortalized cells under two-photon illumination.

charges (*Figure 1B*) met two of our key design criteria: improved voltage responsiveness in HEK293A cells (*Figure 1—figure supplement 1I*) and a slower off-rate than ASAP1 (*Table 1*). We thus designated this more sensitive variant ASAP2s.

Prior to deploying ASAP2s in brain tissue, we first characterized its performance using immortalized cells at 22°C. In response to a 1 s, 100 mV depolarization from –70 mV in HEK293A cells, ASAP2s exhibited a steady-state fluorescence change of –38.7 ± 1.1%, versus –23.3 ± 1.1% for

**Table 1.** Response kinetics of ASAP indicators and ArcLight Q239 in HEK293A cells at 22°C.

| | ASAP1 | ASAP2s | ArcLight Q239 |
|---|---|---|---|
| Depolarization (–70 to 30 mV) | | | |
| $\tau_{fast}$ (ms) | 2.9 ± 0.3 | 5.2 ± 0.4 | 20 ± 2 |
| $\tau_{slow}$ (ms) | 161 ± 33 | 63 ± 11 | 267 ± 13 |
| % fast | 74 ± 5 | 56 ± 7 | 37 ± 7 |
| Repolarization (30 to –70 mV) | | | |
| $\tau_{fast}$ (ms) | 2.3 ± 0.4 | 24 ± 7 | 113 ± 11 |
| $\tau_{slow}$ (ms) | 177 ± 38 | 106 ± 47 | 367 ± 32 |
| % fast | 63 ± 6 | 49 ± 17 | 53 ± 8 |
| Hyperpolarization (–70 to –100 mV) | | | |
| $\tau_{fast}$ (ms) | 11 ± 3 | 8.2 ± 0.6 | 20 ± 5 |
| $\tau_{slow}$ (ms) | 131 ± 16 | 104 ± 10 | 208 ± 23 |
| % fast | 59 ± 3 | 53 ± 2 | 49 ± 3 |
| Repolarization (–100 to –70 mV) | | | |
| $\tau_{fast}$ (ms) | 15 ± 3 | 13 ± 1 | 42 ± 12 |
| $\tau_{slow}$ (ms) | 131 ± 14 | 114 ± 10 | 265 ± 75 |
| % fast | 52 ± 2 | 51 ± 1 | 57 ± 3 |

n = 4–5 cells per sensor. Data are presented as mean ± SEM.

ASAP1 (mean ± standard error of the mean, p<0.001, t-test), a 66% improvement (*Figure 1C,D*). The steady-state response of ASAP2s to voltage steps has the largest amplitude among GEVIs based on fluorescent proteins, surpassing ArcLight Q239 (−32 to −35%, [*Jin et al., 2012*; *Zou et al., 2014*]), MacQ-mCitrine (∼−20%, [*Gong et al., 2014*]), and Ace2N-mNeon (<-5% steady-state, ∼−19% peak, [*Gong et al., 2015*]). We next evaluated the kinetics of ASAP2s and ASAP1 by fitting the optical responses to 100 mV step depolarizations in HEK293A cells. We also evaluated ArcLight Q239 (hereby designated as ArcLight), an indicator previously used to benchmark new GEVIs (*Zou et al., 2014*; *St-Pierre et al., 2014*). Fluorescence responses of all GEVIs were best fit by a weighted sum of two time constants (*Table 1*). We focus here on the faster time constants given their greater importance for tracking rapid neuronal activity. The fast depolarization time constants (on-rates) were 5.2 ms for ASAP2s and 2.9 ms for ASAP1 (*Table 1*), much faster than the 20 ms for ArcLight. Critically, ASAP2s' fast response to repolarization (fast off-rate) was 24 ms or ∼10 fold slower than ASAP1 (*Table 1*), a useful change for improving spike detection (*Wilt et al., 2013*), while being still sufficiently rapid to track fast trains of AP waveforms at 100 Hz in single trials (*Figure 1— figure supplement 1J*).

We next confirmed that the R415Q mutation in ASAP1 did not affect its brightness under both one- and two-photon excitation. As brightness depends on the membrane potential, we evaluated indicator brightness in HEK293-Kir2.1, a cell line with plasma membrane potential around –77 mV, close to the resting membrane potential of many neurons (*Zhang et al., 2009*). When expressed in these cells, ASAP2s matched the brightness of ASAP1 under both one- and two-photon illumination (*Figure 1—figure supplement 2A,B*). Two-photon microscopy experiments were performed by exciting GEVIs at ∼920 nm, the wavelength at which ASAP1 and ASAP2s are maximally excited (*Figure 1E*). The commonly-used fluorescent protein EGFP, which we evaluated as a standard, also produced maximal brightness when excited at 920 nm, consistent with previous reports (*Drobizhev et al., 2011*).

We next compared the photostability of ASAP1 and ASAP2s in HEK293-Kir2.1 cells. We also sought to compare the photobleaching kinetics of the ASAP indicators to those of ArcLight and EGFP. Because of fundamental differences in apparent photobleaching between membrane and cytoplasmic probes (*Brinks et al., 2015*), it is most appropriate to compare photostability under conditions where EGFP is membrane localized. We therefore created a standard by replacing the

circularly permuted GFP in ASAP1 with EGFP; we designated the resulting probe ASAP1::EGFP. Under one-photon widefield microscopy at identical illumination power, ASAP1 and ASAP2s photobleached similarly to each other and to ArcLight and more slowly than ASAP1::EGFP (*Figure 1—figure supplement 3A,B*). Consistent with the photobleaching behavior of superfolder GFP, the fluorescent protein from which the GFP in ASAP indicators was derived, photobleaching kinetics were best fit by a two-term exponential (*Pédelacq et al., 2006*). In contrast, ArcLight and ASAP1::EGFP photobleached with monoexponential kinetics. We observed a partial recovery in ASAP1 and ASAP2s fluorescence following incubation in darkness (*Figure 1—figure supplement 3C–F*), possibly due to dark reversion of bleached to unbleached molecules (*Sinnecker et al., 2005*), diffusion of unbleached probes to the illuminated area, or both (*Pincet et al., 2016*). Longer dark incubation resulted in greater fluorescence recovery, yielding 8.9 ± 0.6% recovery of the original fluorescence for a 0.5 min incubation, and 22.4 ± 0.7% for a 5-min incubation (*Figure 1—figure supplement 3C, D*). The magnitude of fluorescence recovery in dark-incubated ASAP1- and ASAP2s-expressing cells was similar (*Figure 1—figure supplement 3E,F*).

Under two-photon laser scanning excitation, ASAP1 and ASAP2s rapidly lost ~30% of their initial brightness but then bleached at slower or similar rates compared with ArcLight and ASAP1::EGFP (*Figure 1—figure supplement 4A,B*). The photobleaching kinetics of all probes was best fit using exponentials with an additional term compared with their one-photon photobleaching curves. We do not have a mechanistic explanation for this difference in photobleaching kinetics. Increasing the power per pixel increased the rate of photobleaching, but the relationship between photobleaching kinetics and laser power was complex and nonlinear (*Figure 1—figure supplement 4C,D*). Photobleaching rates are expected to vary based on the power at each pixel, which depend not only on laser power but also on the pixel duty cycle, defined as the percentage of time the laser is exciting each location corresponding to the image pixels. The pixel duty cycle is calculated as the product of the frame acquisition rate and the length of time the laser resides at each location in each sweep (the dwell time). As anticipated, two conditions with identical power per pixel gave similar photobleaching rates, despite differing in laser power, frame rate, and dwell time (*Figure 1—figure supplement 4E*). Incubation in the dark resulted in fluorescence recovery, as observed under one-photon illumination: 20–25% of original fluorescence was recovered following a 5-min dark incubation for both ASAP1 and ASAP2s and across two conditions with distinct laser power (*Figure 1—figure supplement 4F,G*). If this reversible photobleaching could be repeatedly obtained over multiple cycles, it would provide a strategy for mitigating photobleaching, for example in longitudinal studies where the same group of cells are repeatedly imaged over multiple days or months.

## Benchmarking GEVIs for imaging voltage dynamics in cardiomyocytes and neurons

We next sought to compare the ability of ASAP2s and other indicators to report voltage dynamics in excitable cells. Given the interest in using indicators to image cardiac electrical activity (*Kaestner et al., 2015*), we first expressed our indicators in cardiomyocytes differentiated from human embryonic stem cells. Consistent with our results in HEK293A cells, the response amplitude of ASAP2s to cardiac potentials was greater than that of the other indicators, reaching −45.1 ± 1.5% compared with −24.0 ± 1.8% for ASAP1 and −32.9 ± 1.8% for ArcLight (*Figure 2A–D*, *Video 1*). In cardiomyocytes derived from induced pluripotent stem cells, the response amplitude of ASAP2s to action potentials was −29.1 ± 2.1% compared with −16.8 ± 2.2% for ASAP1 and −26.3 ± 2.8% for ArcLight. The faster kinetics of the ASAP indicators compared to ArcLight enabled a shorter time to peak when reporting cardiac action potentials (*Figure 2E–F*, *Figure 2—figure supplement 1*). These results demonstrate that ASAP2s can image cardiac action potentials, as previously shown with other voltage indicators in vitro (*Kaestner et al., 2015*; *Tian et al., 2011*; *Leyton-Mange et al., 2014*; *Chang Liao et al., 2015*; *Werley et al., 2017*) and in vivo (*Chang Liao et al., 2015*; *Tsutsui et al., 2010*; *Hou et al., 2014*).

Before examining the ability of ASAP2s to report neuronal voltage signals, we first evaluated its membrane localization. Proper plasma membrane localization is crucial for detecting voltage events, as GEVIs trapped in internal membrane structures or aggregates do not respond to neuronal activity but can be brightly fluorescent (*Baker et al., 2007*). Such bright yet inactive GEVI molecules would thus reduce the overall cellular fluorescence response. Moreover, the response amplitude would show large variation between subcellular locations depending on the proportion of properly

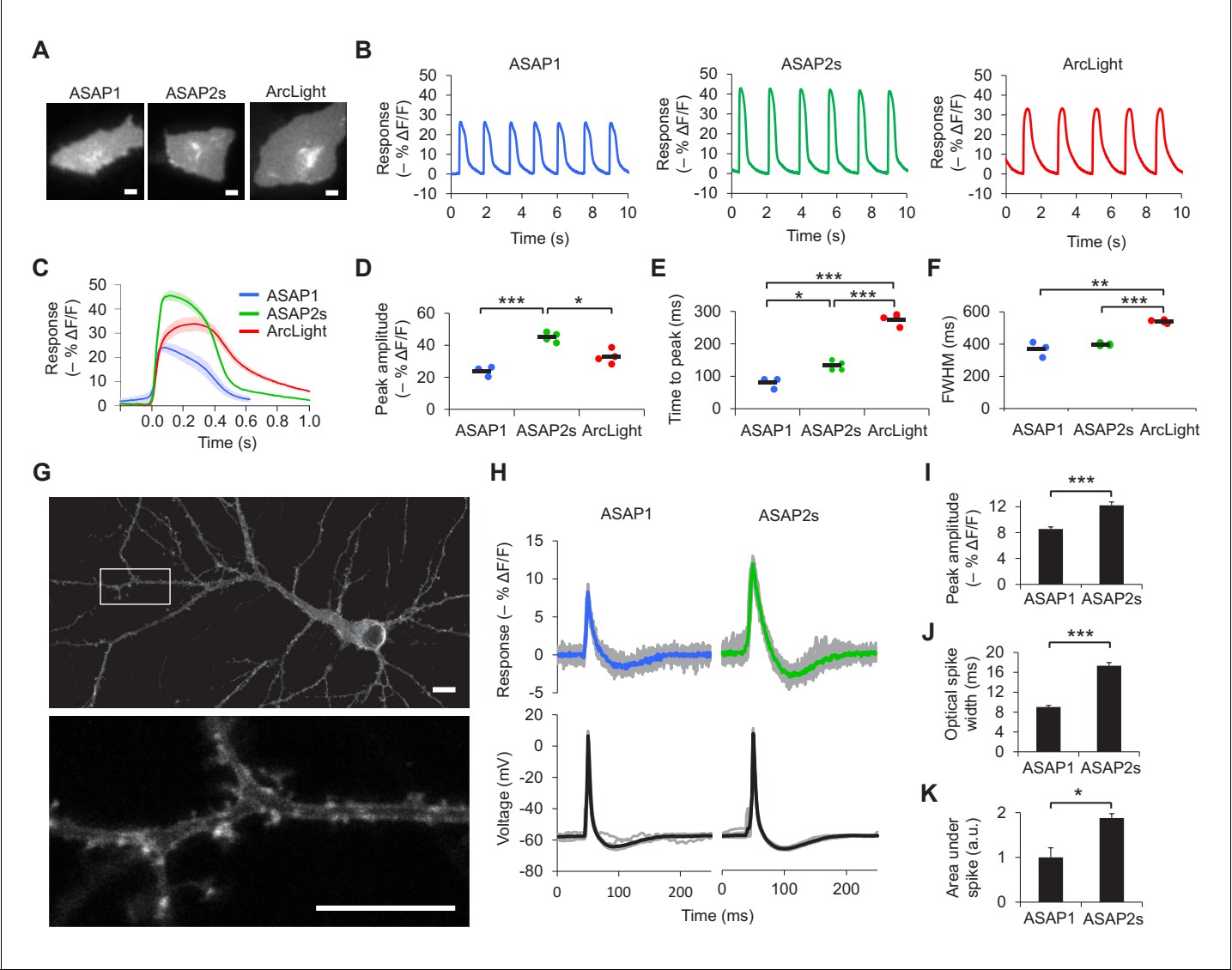

**Figure 2.** Characterization of ASAP2s in cardiomyocytes and neurons. (**A**) Representative human embryonic stem-cell-derived cardiomyocytes (hESC-CMs) expressing ASAP1, ASAP2s, or ArcLight. Scale bar, 10 μm. (**B**) Representative single-trial responses to spontaneous cardiac APs from the cells in panel A. (**C**) Mean fluorescence response to spontaneous APs from hESC-CMs expressing ASAP1 (blue, n = 3 cells), ASAP2s (green, n = 4), or ArcLight (black, n = 4), with 5-15 APs per cell. Shaded areas are ±1 SEM. Prior to averaging, baseline fluorescence was set to the minimum fluorescence value. Because hESC-CMs show variability in beating rate, averaging was performed in a time window that contains only one AP across all cells analyzed. As this window does not include the lowest point of the cardiac cycle in all cells, average traces do not all start at or return to baseline in this window. The same cells were further analyzed in panels D to F. (**D**) Maximal fluorescence response of ASAP1, ASAP2s, or ArcLight to cardiac APs. (**E**) Time from the start of the fast-rising phase of the AP to the fluorescence peak. (**F**) Full-width at half-maximum (FWHM) of the fluorescence trace. (**G**) ASAP2s fluorescence is well localized at the plasma membrane in a representative cultured rat cortical neuron imaged by confocal microscopy. Bottom, magnified image of the boxed regions. Scale bar, 10 μm. (**H**) Responses of ASAP1 and ASAP2s to APs in representative cultured hippocampal neurons. Thick traces are mean responses over all APs for a given representative neuron. Single-trial traces are shown in gray (n = 11 APs per neuron), with single examples shown in *Figure 2—figure supplement 3A*. (**I**) Mean peak response to current-triggered APs in cultured hippocampal neurons. n = 8 (ASAP1) and 5 (ASAP2s) neurons. For each neuron, we measured 2 to 25 APs (n = 118 total APs for ASAP1, n = 56 for ASAP2s). AP peak voltage was 9.5 ± 2.0 mV for ASAP1 and 15.2 ± 2.0 mV for ASAP2s (mean ± SEM). (**J**) Widths of the optical spikes at half-maximal height. AP widths in the voltage traces were 4.2 ± 0.2 and 4.6 ± 0.1 ms for ASAP1 and ASAP2s, respectively (mean ± SEM). The data is from the same neurons as panel I. (**K**) Areas under the curve of fluorescence responses. The data is from the same neurons as panel I. For all panels: comparisons were performed using the t-test with Bonferroni correction for multiple comparisons, except for panel I, where the Mann-Whitney U-test was used. *p<0.05, **p<0.01, ***p<0.001. DOI: 10.7554/eLife.25690.007

The following figure supplements are available for figure 2:

**Figure supplement 1.** Voltage imaging of cardiomyocytes derived from induced pluripotent stem cells.

*Figure 2 continued on next page*

*Figure 2 continued*

**Figure supplement 2.** Plasma membrane localization of the ASAP indicators in cultured hippocampal neurons.

**Figure supplement 3.** Characterization of ASAP2s in cultured hippocampal neurons.

folded and localized GEVI molecules, complicating the study of how voltage transients propagate in single neurons. We observed that ASAP2s localized efficiently to the plasma membrane in cultured neurons (*Figure 2G*, *Figure 2—figure supplement 2*), as previously reported with ASAP1 (*St-Pierre et al., 2014*). Applied voltage steps elicited ASAP2s fluorescence changes, without apparent non-responding clusters or aggregates (*Video 2*).

We then measured responses to single APs in cultured neurons. We predicted that the larger response amplitude of ASAP2s, coupled with its slower inactivation rate, would result in larger and longer responses to single APs. Indeed, in neuronal cultures, ASAP2s reported APs with a $-12.2 \pm 0.5\%$ fluorescence change, compared with $-8.6 \pm 0.3\%$ for ASAP1, a 42% improvement in response amplitude (*Figure 2H,I*, *Figure 2—figure supplement 3A*). ASAP2s' responses were also longer in duration (*Figure 2J*), consistent with its slower kinetics of repolarization. The combination of higher response amplitude and longer response duration resulted in a 90% greater integrated fluorescence change with ASAP2s than with ASAP1 (*Figure 2K*). Under widefield illumination of these neurons, ASAP2s photobleached similarly to ASAP1 (*Figure 2—figure supplement 3B,C*).

## Benchmarking GEVIs for two-photon microscopy in *Drosophila*

Having optimized and characterized the ASAP indicators in culture, we next sought to benchmark the performance of ASAP2s for in vivo detection of voltage dynamics using two-photon microscopy. The *Drosophila* visual system has recently emerged as a useful platform for evaluating voltage indicators in vivo (*Yang et al., 2016*), as it is accessible for imaging, visual stimuli can be presented with temporal and spatial precision, and many cell types are well described and can be genetically targeted. Two-photon imaging is especially critical for monitoring voltage dynamics in the fly visual system, as it uses infrared light that does not excite fly photoreceptors, unlike the visible spectrum wavelengths commonly used for one-photon microscopy of most biosensors (*Salcedo et al., 1999*).

We tested several GEVIs in the fruit fly visual system by expressing them selectively in L2 neurons. Along with L1 and L3, L2 neurons are monopolar cells of the lamina that receive direct inputs from the R1-6 photoreceptors (*Figure 3A*) (*Meinertzhagen and O'Neil, 1991*; *Sanes and Zipursky, 2010*). L1-3 each retinotopically tile the visual field and provide critical outputs to the medulla, the next layer in visual processing. We positioned awake flies in front of a screen displaying visual stimuli (alternating 300 ms dark and light flashes) and imaged L2 axon terminals through a window cut in the cuticle at the back of the head (*Figure 3A*).

The fluorescence responses of both ASAP1 and ASAP2s indicated that L2 axon terminals

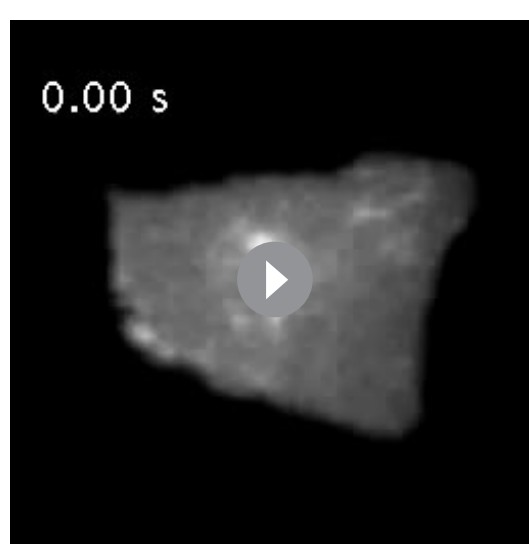

**Video 1.** ASAP2s optical response to cardiac APs in a human embryonic stem cell-derived cardiomyocyte (hESC-CM). A hESC-CM was transfected with ASAP2s at 27 days post-differentiation and was imaged three days later at 100 Hz and with a power density of 11 mW/mm$^2$ at the sample plane. Quantification of the fluorescence response during the first 10 s is shown in *Figure 2B*. The movie corresponds to a single trial without filtering, smoothing, or photobleaching correction.

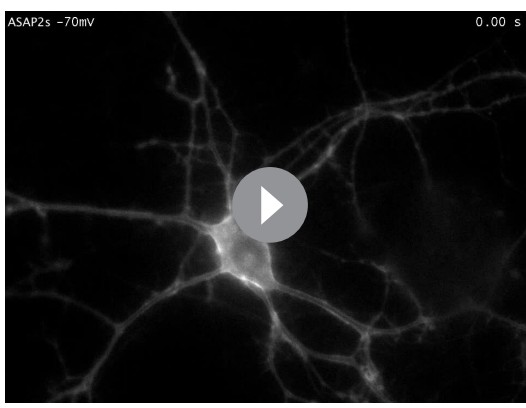

**Video 2.** ASAP2s responses to step voltages in a patch-clamped cultured hippocampal neuron. ASAP2s fluorescence was captured while its transmembrane voltage was stepped from –70 to 0, 30, or 50 mV as labelled. Frames were acquired at 20 Hz and played back in real time. The movie corresponds to a single trial without filtering, smoothing, or photobleaching correction.

transiently depolarize to light decrements and transiently hyperpolarize to light increments (*Figure 3B*), consistent with electrophysiological recordings in lamina monopolar cells (*Zettler and Järvilehto, 1971*) and our prior GEVI-imaging experiments (*Yang et al., 2016*). In line with our in vitro observations, ASAP2s produced significantly larger mean responses than ASAP1 (a 37% increase in amplitude for depolarizations and a 39% increase in amplitude for hyperpolarizations) but was slightly slower (*Figure 3C,D*). Similarly, these performance metrics show that ASAP2s produces larger but slower responses than ASAP2f, a recent voltage indicator primarily characterized for applications in flies (*Yang et al., 2016*). For both ASAP1 and ASAP2s, we could also observe single-cell and single-trial responses, although there was trial-to-trial variability (*Figure 3B*).

Because ArcLight has been previously used in flies (*Cao et al., 2013*; *Sitaraman et al., 2015*), we also characterized this sensor under identical imaging conditions. We found that ArcLight produced fluorescence responses with amplitudes similar to ASAP1 but smaller than ASAP2s (*Figure 3B,D*). The kinetics of ArcLight fluorescence changes were significantly slower than those of both ASAP2s and ASAP1 (*Figure 3D*), in agreement with their respective response kinetics in cultured cells (*Table 1*, *Figure 2B,C,E,F*, *Figure 2—figure supplement 1*). This result is also consistent with previous findings using ArcLight in the fly with one-photon microscopy in which its fluorescence traces had peaks wider than the underlying voltage signals (*Cao et al., 2013*). Some GEVIs based on the same voltage-sensing domain as ArcLight exhibit faster kinetics. However, they exhibit smaller response amplitudes; they have not yet been tested in flies; and the vast majority have not been evaluated under two-photon illumination (*Akemann et al., 2012*; *Barnett et al., 2012*; *Baker et al., 2012*; *Mishina et al., 2012*; *Han et al., 2013*; *Tsutsui et al., 2013*; *Mishina et al., 2014*; *Piao et al., 2015*; *Treger et al., 2015*).

Over 30 min of continuous illumination, the photostability of ASAP1, ASAP2s, and ArcLight was comparable, with all three indicators bleaching to ~30% of their initial brightness (*Figure 3—figure supplement 1A,B*). ASAP1 and ASAP2s both rapidly bleached by ~25% within 2 s, consistent with the photobleaching characteristics of a close variant of the GFP used in those two indicators (*Pédelacq et al., 2006*). ArcLight did not exhibit this rapid bleaching, but its brightness decreased more rapidly than ASAP indicators after these initial 2 s. All three indicators enabled imaging of L2 voltage dynamics with a high signal-to-noise ratio (SNR) for over 30 min (*Figure 3—figure supplement 1C*).

To extend our comparisons to GEVIs that use opsin domains for voltage sensing, we also evaluated Ace2N-2AA-mNeon (*Gong et al., 2015*), an indicator previously used in flies under one-photon illumination. We also tested MacQ-mCitrine (*Gong et al., 2014*) as a second example of an opsin-based indicator. Both indicators fluoresced brightly with two-photon excitation in L2 (*Figure 3—figure supplement 2A,B*), but produced minimal (Ace2N-2AA-mNeon) or undetectable (MacQ-mCitrine) responses to visual stimulation (*Figure 3B*). Unexpectedly, the response polarity of Ace2N-2AA-mNeon was inverted compared to its response under one-photon illumination (*Gong et al., 2015*), an observation robust to excitation wavelength between 920 and 1010 nm (*Figure 3—figure supplement 2C,D*). Overall, the poor responses of both indicators in this in vivo benchmark are consistent with published observations that voltage responses of opsin-based GEVIs can be greatly diminished under two-photon excitation (*Brinks et al., 2015*).

Finally, to determine whether calcium imaging could provide the same information as voltage imaging in L2 neurons, we also imaged calcium dynamics using a genetically encoded calcium indicator. We chose GCaMP6f given that it features dramatically improved kinetics over previous

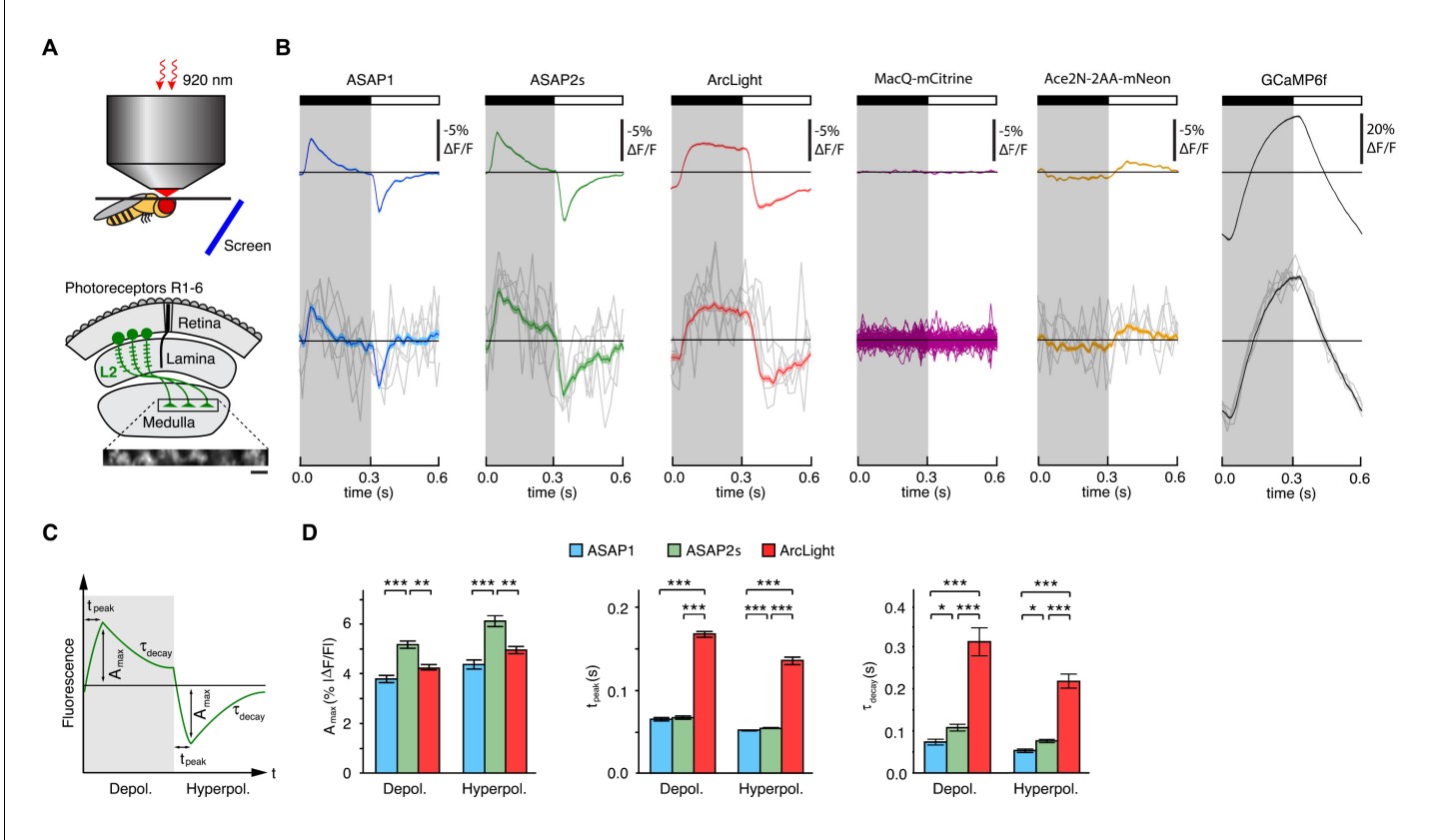

**Figure 3.** Two-photon imaging of subcellular voltage responses to physiological stimuli in *Drosophila*. (**A**) Schematic illustration of the imaging setup (top) and the fruit fly visual system around L2 cells (bottom). Inset, example of the region imaged, with six L2 terminals expressing ASAP2s. Expression of the other sensors was comparable. Scale bar, 5 μm. (**B**) L2 responses to alternating 300 ms dark and light flashes, as measured with different indicators. The black bar indicates the dark period and the white bar indicates the light period. Top, mean response across all cells (n = 44 cells from 3 flies for ASAP1, 111 cells from 5 flies for ASAP2s, 65 cells from 5 flies for ArcLight, 64 cells from 3 flies for MacQ-mCitrine, 23 cells from 4 flies for Ace2N-2AA-mNeon, and 232 cells from 10 flies for GCaMP6f). Each cell contributes its average response across 100 trials (one trial = 1 dark flash and one light flash). Horizontal black lines are the mean fluorescence per trial; indicators with slow or asymmetric responses to dark and light flashes can produce traces that do not begin or end at the mean fluorescence. Bottom, for ASAP1, ASAP2s, ArcLight, Ace2N-2AA-mNeon, and GCaMP6f, five exemplar single-trial responses from a single representative L2 cell (gray) and the same cell's mean response over all trials (colored, n = 100 trials). Solid line is mean; lighter shading is ±1 SEM. Because MacQ-mCitrine didn't produce an apparent stimulus-evoked response when averaging across all cells (top trace), we plotted the mean optical traces of individual cells (bottom traces) to illustrate that stimulus-evoked responses were also not apparent in any cells. (**C**) Schematic illustrating the response parameters quantified in panel D. $A_{max}$, the maximal amplitude of the fractional fluorescence change ($|\Delta F/F|$); $t_{peak}$, the time at which $A_{max}$ occurs, relative to the start of the flash; $\tau_{decay}$, the time constant of the decay from $A_{max}$. (**D**) For each voltage sensor, $A_{max}$, $t_{peak}$, and $\tau_{decay}$ are plotted for depolarizations (left) and hyperpolarizations (right). Sample sizes are the same as in panel B, except for some measurements of $\tau_{decay}$ that did not meet our inclusion criterion (see Materials and methods). $A_{max}$ and $\tau_{decay}$ were analyzed with the t-test, while $t_{peak}$ was analyzed with the Mann-Whitney U-test. We used the Bonferroni method to correct for multiple pairwise comparisons between ASAP1, ASAP2s, and ArcLight values for a given response parameter ($A_{max}$, $t_{peak}$, and $\tau_{decay}$) and sign of voltage change (depolarization or hyperpolarization). *p<0.05, **p<0.01, ***p<0.001.

The following figure supplements are available for figure 3:

**Figure supplement 1.** Photostability of ASAP1, ASAP2s, and ArcLight in *Drosophila* L2 axon terminals under two-photon illumination.

**Figure supplement 2.** Expression and responses of FRET-opsin voltage indicators in *Drosophila* L2 axon terminals.

calcium indicators (*Chen et al., 2013*) and is therefore better suited for monitoring the 300-ms light flashes used here. The peak response amplitude of GCaMP6f was substantially larger than that of any of the voltage sensors, and its single-cell and single-trial responses had correspondingly higher SNR (*Figure 3B*). However, the GCaMP6f response was not transient, continuing to rise or fall during

the entire duration of the flash. This result is consistent with previous studies of L2 using the calcium indicator TN-XXL (*Reiff et al., 2010*; *Clark et al., 2011*). Importantly, the observed calcium indicator traces do not correlate with GEVI traces in a simple manner, illustrating the difficulty of using calcium imaging to infer voltage dynamics. Overall, our results demonstrate the suitability of the ASAP sensors for two-photon imaging of voltage dynamics in vivo, with ASAP2s providing the best balance of large response amplitude and sufficiently fast kinetics.

## Random-access, single-trial two-photon imaging of action potentials in organotypic slice cultures

Having benchmarked ASAP indicators in vitro and in vivo, we next sought to determine whether ASAP-family sensors could enable single-trial detection of rapid voltage transients in single cells in organotypic slice cultures under two-photon illumination. We chose to perform voltage detection using random-access multiphoton microscopy (RAMP, *Figure 4A*), a technique for fast imaging of arbitrary locations in two- or three-dimensional space by rapid movement of the laser beam (*Lechleiter et al., 2002*; *Iyer et al., 2006*; *Duemani Reddy et al., 2008*). RAMP is a useful technique for rapid imaging of multiple neurons or subcellular locations within a single neuron. Therefore, imaging ASAP indicators with RAMP could provide the means to record voltage at multiple sites with exquisite spatial precision and temporal resolution, a combination we term Fast Excitation of Voltage Indicators by RAMP, or FEVIR.

We expressed ASAP2s and ASAP1 in organotypic hippocampal slice cultures and imaged fluorescence generated by two-photon illumination. All experiments were performed at room temperature (22°C). We observed that ASAP2s expressed well along neuronal membranes (*Figure 4B*) and that the photon emission from ASAP2s was comparable to that of ASAP1 (*Figure 4C*). We determined that the resting membrane potential, membrane capacitance, and input resistance of GEVI-expressing and untransfected neurons were statistically indistinguishable (*Figure 4—figure supplement 1*), consistent with a previous observation with ASAP1 (*St-Pierre et al., 2014*).

We first tested the ability of FEVIR to detect evoked APs. Both ASAPs detected APs, with ASAP2s producing a fluorescence change of −15.0 ± 0.6%, a 79% improvement over the −8.4 ± 0.5% response amplitude when using ASAP1 (*Figure 4D,E*). These two-photon response amplitudes are remarkably similar to one-photon responses, consistent with previous observations with ASAP1 (*Brinks et al., 2015*) and confirming that the larger response of ASAP2s over ASAP1 observed under one-photon excitation was preserved under two-photon illumination. We evaluated ASAP2s further and demonstrated it can report spontaneous APs (*Figure 4F*, *Figure 4—figure supplement 2*). Both ASAP2s and ASAP1 could also reliably detect subthreshold depolarization and hyperpolarization waveforms in single trials (*Figure 5*).

To quantify GEVIs' ability to detect spikes, we calculated the detectability metric d', which provides a measure of our ability to distinguish a spike from noise in idealized conditions of well-isolated spikes occurring on an otherwise stable membrane potential (*Wilt et al., 2013*). For neurons imaged at 925 Hz, we obtained a d' value of 53.9 ± 3.2 (n = 23 cells) for ASAP2s, much larger than 15.0 ± 3.7 (n = 15 cells) for ASAP1 (p<0.0001, Mann-Whitney U test). The d' value for both ASAP1 and ASAP2s indicate a greater than 99% detection rate and a false-positive rate per frame of less than $10^{-8}$ for both ASAP1 and ASAP2s (*Wilt et al., 2013*). Actual detectability will vary in practice depending on other variables such as illumination power, GEVI expression levels, imaging depth, indicator photobleaching, spike rate, and the presence of subthreshold depolarizations.

We characterized the kinetics of the ASAP indicators in our system by oversampling at 3700 Hz (*Figure 6A–E*). ASAP1 exhibited AP-induced transients with a mean time-to-peak of 3.1 ms (*Figure 6B,C*) and a duration (full-width at half-maximum) of 3.8 ms (*Figure 6D,E*). As predicted from its kinetics in cultured cells (*Table 1*), ASAP2s kinetics were slower, with a time-to-peak of 6.0 ms and a duration of 21.1 ms. ASAP2s' photobleaching was best fit with both fast and slow exponentials (*Figure 6—figure supplement 1*), as previously observed under other illumination and expression conditions (*Figure 1—figure supplement 4*).

We next explored how changing the scanning frequency could optimize the SNR. ASAP2s signal amplitudes remained relatively constant across all frequencies tested (*Figure 7A*), consistent with the longer durations of its optical transients. Noise decreased at lower sampling frequencies, as expected from the collection of more photons per time point (*Figure 7B*). Because of its higher

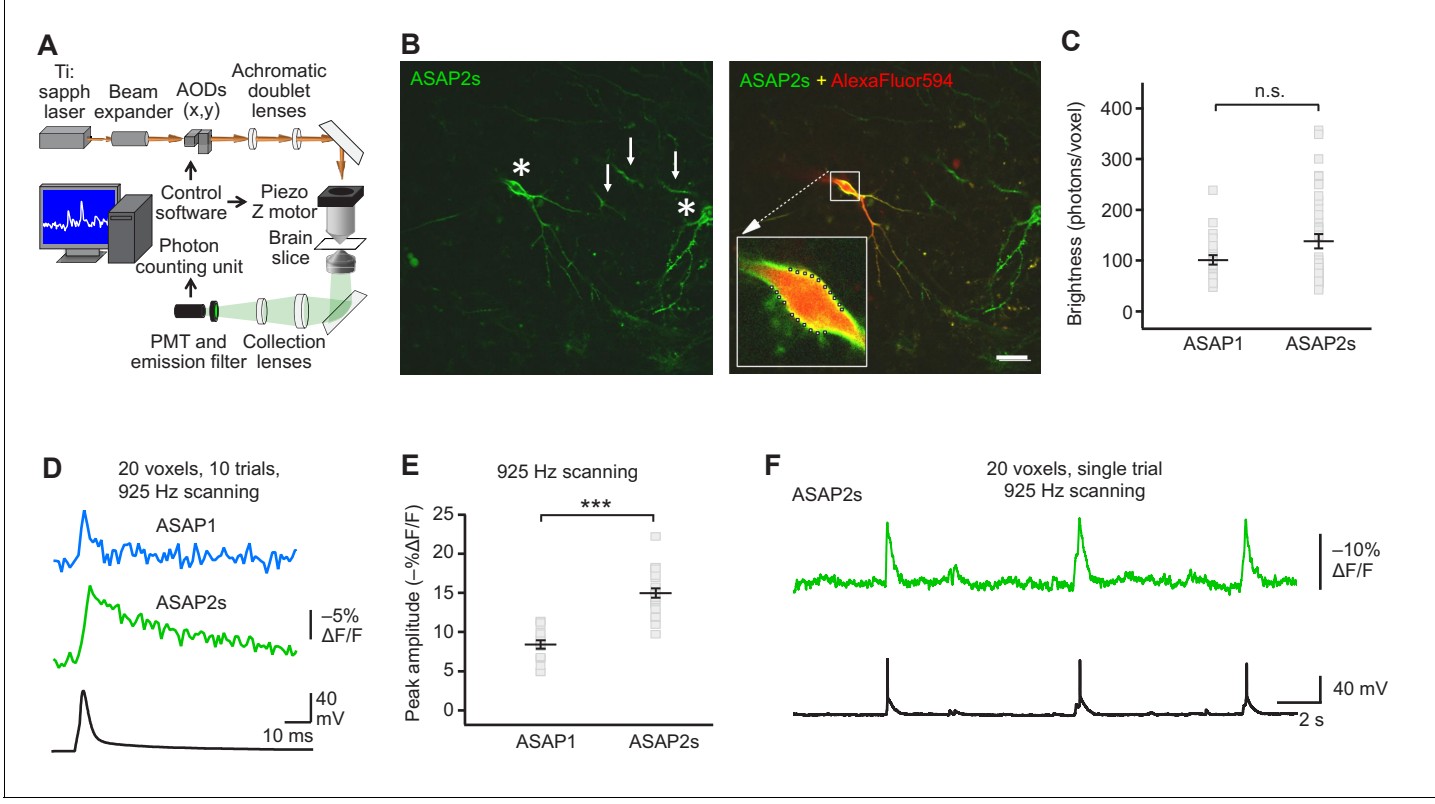

**Figure 4.** FEVIR: fast random-access two-photon imaging of GEVI responses in organotypic hippocampal slice cultures. (A) Schematic of our random-access multi-photon (RAMP) imaging system. (B) Representative two-photon single-plane image of ASAP2s-expressing neurons in an organotypic hippocampal slice culture. Left, two neurons can be seen (asterisks), together with processes belonging to ASAP2s-expressing neurons in different planes (arrows). Right, overlay of ASAP2s and AlexaFluor594 fluorescence identifies the neuron recorded in whole-cell configuration. Inset shows how 20 individual voxels (squares) can be selected along the plasma membrane when imaging at 925 Hz. Scale bar, 20 µm. (C) Mean number of photons emitted per voxel during a 50 µs exposure (n = 23 neurons for ASAP1, n = 40 for ASAP2s). Data obtained at a holding potential of –70 mV. Individual neurons are shown as gray squares. Black horizontal bars are the means, and error bars are the SEM (p>0.05, Mann-Whitney U-test). (D) Representative ASAP2s and ASAP1 responses to a single current-evoked AP (black trace). Optical recordings were acquired at 925 Hz with 20 voxels per neuron. Traces are the average of 10 trials. (E) Peak response amplitudes at the soma induced by single current-triggered APs were significantly larger with ASAP2s (n = 17 neurons for ASAP1, n = 23 for ASAP2s, ***p<0.001, t-test). Each data point (gray squares) corresponds to the mean response amplitude to 10 APs per neuron. Black horizontal bars are the means, and error bars are the SEM. (F) Representative single-trial ASAP2s response to spontaneous APs. The voltage trace (bottom) was obtained by simultaneous patch clamping.

The following figure supplements are available for figure 4:

**Figure supplement 1.** Plasma membrane excitability of the ASAP indicators.

**Figure supplement 2.** Detecting spontaneous APs using ASAP2s.

**Figure supplement 3.** Detecting evoked APs using ASAP2f.

**Figure supplement 4.** Plasma membrane localization and RAMP imaging of Ace2N-4AA-mNeon in organotypic hippocampal slice cultures.

response amplitude, ASAP2s showed higher SNR than ASAP1 at all frequencies (*Figure 7C*). ASAP2s' SNR increased with decreasing scanning frequencies, reaching 10.0 ± 1.6 at 231 Hz from 5.4 ± 0.4 at 3700 Hz. In contrast, ASAP1's SNR remained relatively constant at ~5 across all frequencies; given the short durations of ASAP1 transients in response to APs, lower frequencies reduce noise but also reduce response amplitude (*Figure 7*). Overall, these results show that ASAP2s and sub-kilohertz scanning frequencies provide the best SNR for detection of low-frequency evoked APs when performing FEVIR at room temperature in our experimental preparation.

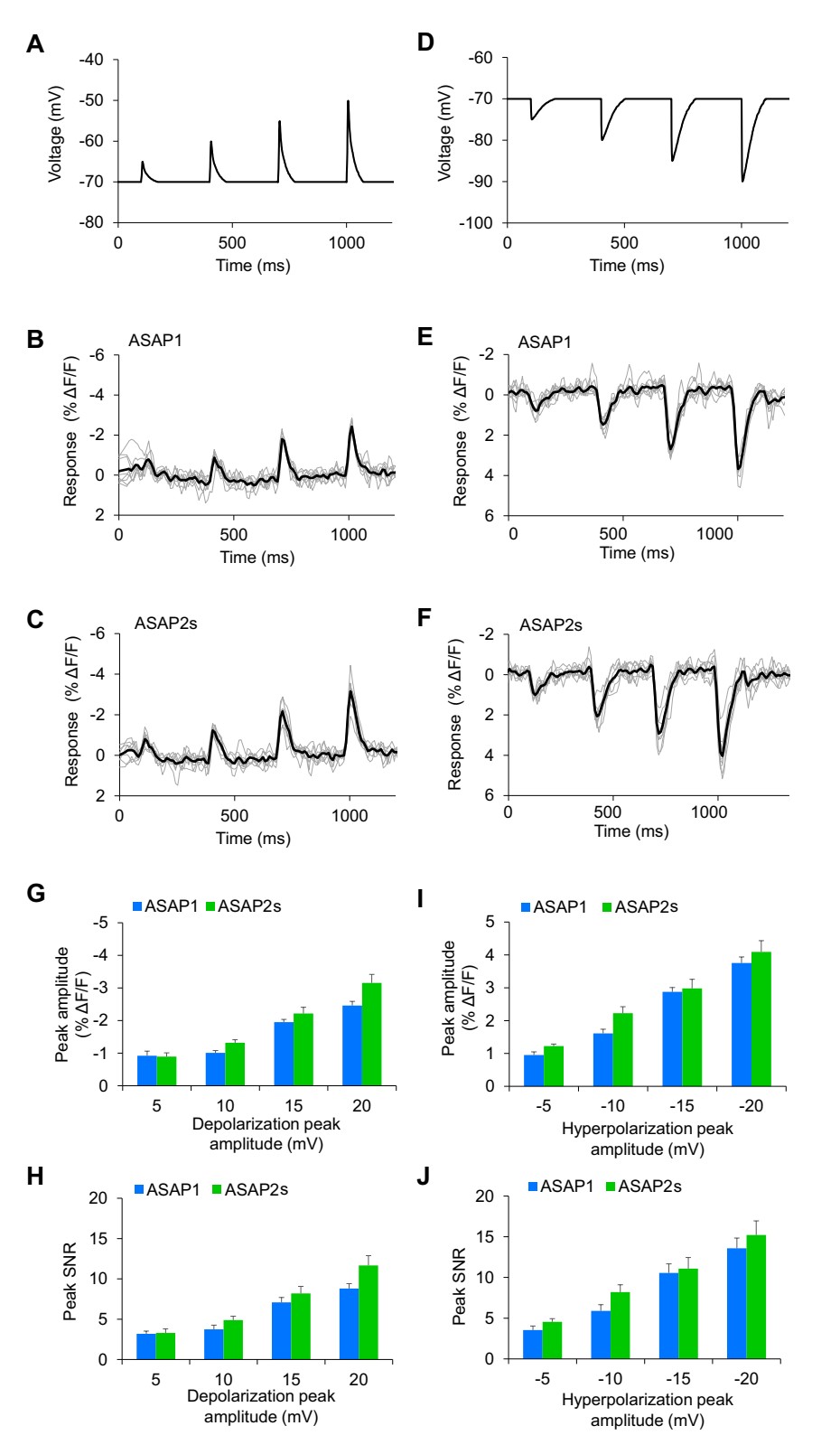

**Figure 5.** Detecting subthreshold depolarizations and hyperpolarizations in single trials. (A–F) ASAP1 and ASAP2s can detect subthreshold potential and hyperpolarization waveforms in single trials in organotypic hippocampal slice cultures. Optical recordings were acquired at 462 Hz. (A) Subthreshold depolarization waveforms had peak amplitudes of 5, 10, 15, and 20 mV; a time-to-peak of 8 ms; and full width at half maximum of 14.5 ms. (B,C) Responses to subthreshold depolarization waveforms using ASAP1 (B) and ASAP2s (C). The mean response is shown in black, and single-trial

*Figure 5 continued on next page*

*Figure 5 continued*

responses are in gray (n = 8 neurons per indicator). (**D**) Hyperpolarization waveforms had peak amplitudes of −5,–10, −15, and −20 mV; a time-to-peak of 5 ms; and full width at half maximum of 39 ms. (**E,F**) Responses to hyperpolarization waveforms using ASAP1 (**E**) and ASAP2s (**F**). The mean response is shown in black, and single-trial responses are in gray (n = 8 neurons per indicator). For panels B,C,E,F, raw traces were smoothed with a window size of 10 time points. (**G,H**) Quantification of the fluorescence response amplitudes (**G**) and SNR (**H**) to subthreshold depolarizations. (**I,J**) Quantification of the fluorescence response amplitudes (**I**) and SNR (**J**) to hyperpolarizations. Differences in peak fluorescence responses and SNR between ASAP1 and ASAP2s were not statistically significant (p>0.05, t-test with Bonferroni correction for multiple comparisons).

When active, most neurons fire APs repetitively. Therefore, we next tested whether the ASAP indicators could report high-frequency trains of APs. APs were delivered at 10, 20, 30, and 100 Hz. At higher spike frequencies, the fluorescence did not recover back to baseline between peaks and therefore reduced the response amplitude of subsequent peaks in the train (*Figure 8A–F*). Correspondingly, the detectability (d') of individual peaks in a train decreased as the AP frequency was increased (*Figure 8—figure supplement 1*). The improved sensitivity of ASAP2s balanced its slower kinetics, producing peak responses matching or exceeding those reported with ASAP1 (*Figure 8*). ASAP2s also reported spike trains with larger d' than those reported by ASAP1 for frequencies up to 30 Hz (*Figure 8—figure supplement 1*). For 100-Hz spike trains, the d' values obtained with ASAP1 and ASAP2s were similar to each other and strongly reduced compared to their magnitude at 30 Hz and slower AP firing frequencies.

Finally, we sought to compare ASAP1 and ASAP2s with other recently-reported voltage indicators. Under identical conditions, we observed that the voltage indicator ASAP2f (*Yang et al., 2016*) performed similarly to ASAP1 across all metrics, including brightness (*Figure 4—figure supplement 3*), peak amplitude and SNR when reporting evoked APs (*Figure 4—figure supplement 3*, *Figure 7—figure supplement 1*), kinetics (*Figure 6—figure supplement 2*), and peak response amplitude to individual spikes in AP trains (*Figure 8—figure supplement 2*). We also evaluated indicators of the Ace2N-mNeon family, GEVIs previously reported to detect spikes in brain slices under one-photon illumination with high fidelity (*Gong et al., 2015*). While all cells expressing the variant Ace2N-4AA-mNeon were bright, a significant fraction of the fluorescence was cytoplasmic (*Figure 4—figure supplement 4A*). We performed RAMP microscopy using voxels at the presumed plasma membrane. In response to a 100 mV step depolarization, Ace2N-4AA-mNeon produced a fluorescence change of −1.8 ± 0.4% (*Figure 4—figure supplement 4B–D*). We did not detect an obvious reduction of response amplitude over the course of the step depolarization, in contrast to observations under one-photon illumination (*Gong et al., 2015*). The small response to voltage under two-photon illumination is consistent with our results in flies (*Figure 3B*) and with prior observations with homologous FRET-opsin indicators (*Brinks et al., 2015*). Shortening the linker between the opsin and the fluorescent protein in this indicator may help increase response amplitudes, although this modification can also impair plasma membrane expression (*Gong et al., 2015*). Given the small response of Ace2N-4AA-mNeon to long step voltages under two-photon illumination, we did not evaluate it further.

## Single-voxel, single-trial spike detection with FEVIR in organotypic slice cultures

The results above demonstrate that ASAP-family indicators can report subthreshold depolarizations, hyperpolarizations, and evoked and spontaneous APs in single trials under two-photon microscopy. This is a critical milestone for optical voltage imaging that no GEVI had previously reached. A next milestone, multi-neuron recordings of spontaneous neural activity, will require distributing imaged voxels across neurons. For example, with a laser dwell time of 50 μs and a sampling frequency of 925 Hz, our FEVIR system could image ~20 neurons with one voxel per neuron. We therefore tested whether we could reach the ultimate goal of detecting APs in single trials and single voxels. While ASAP1 responses could not be easily discerned from noise (SNR of 1.1 ± 0.1), ASAP2s responses to single APs in single voxels in single trials were above the mean noise level, giving an SNR of 2.2 ± 0.2 at 925 Hz (*Figure 9A–C*). Similar to ASAP1, single-trial, single-voxel ASAP2f responses were not easily detectable from noise (*Figure 9—figure supplement 1*).

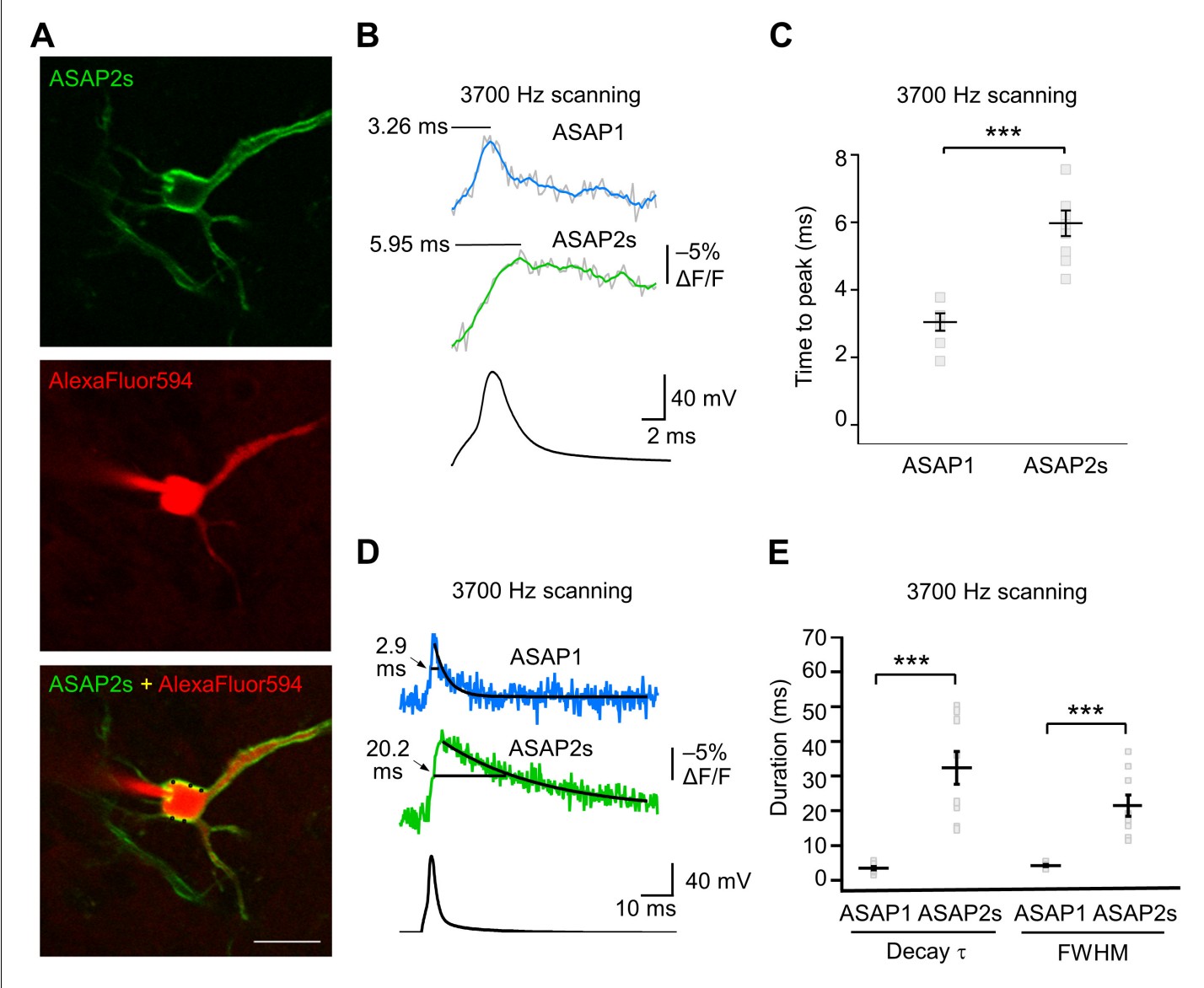

**Figure 6.** GEVI response kinetics to action potentials in organotypic hippocampal slice culture. (**A**) ASAP2s and AlexaFluor594 fluorescence from a representative neuron. Overlay shows the five positions (black squares) selected for imaging at 3700 Hz. Scale bar, 20 µm. (**B**) Representative ASAP1 and ASAP2s responses to current-triggered APs. The indicated time values correspond to the time of the peak response from the beginning of the AP. Optical recordings were acquired at 3700 Hz with five voxels per neuron. The mean of 10 traces is shown (gray trace) along with a five-point moving average (colored traces). The electrophysiological trace was obtained from the same neuron as the ASAP1 optical trace. (**C**) Time-to-peak measured with ASAP1 and ASAP2s (n = 9 neurons for ASAP1, n = 7 for ASAP2s). Black horizontal bars are the means, and error bars are the SEM. ***p<0.001 (t-test). (**D**) Optical spike width (full width at half maximum) of representative ASAP1 and ASAP2s responses to current-triggered APs. The traces are from panel B and shown here over a longer time scale. Decay time constants (τ) were obtained from single-exponential fits. (**E**) Mean decay time constants (τ) and full width at half maximum (FWHM) of ASAP2s and ASAP1 responses to action potentials. n = 7 neurons for ASAP1, n = 9 for ASAP2, ***p<0.001 (t-test).

The following figure supplements are available for figure 6:

**Figure supplement 1.** Photostability of ASAP2s imaged by RAMP microscopy.

**Figure supplement 2.** ASAP2f and ASAP1 report APs in organotypic hippocampal slice cultures with similar kinetics.

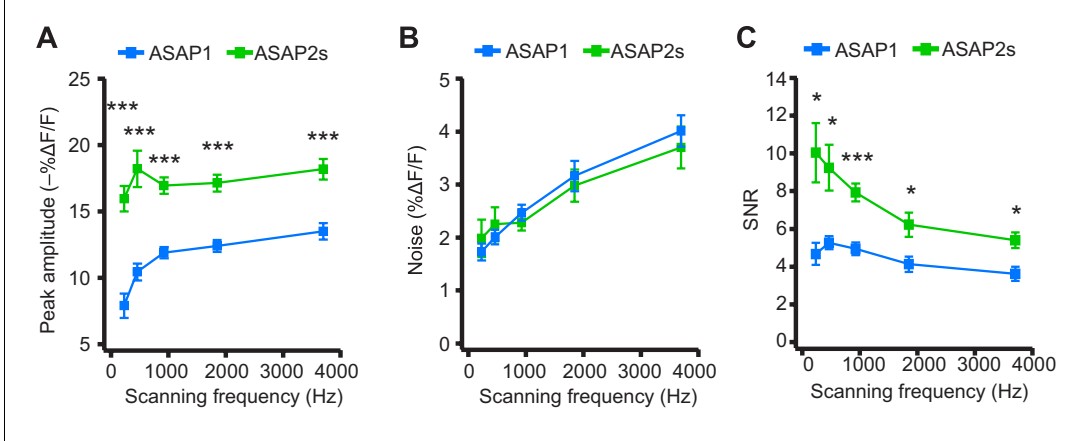

**Figure 7.** Dependence of action potential detection on scanning frequency. (**A**) Peak amplitude of current-evoked APs as a function of the scanning frequency for ASAP2s and ASAP1 in single trials. Voxel dwell time was kept constant at 50 µs, and the number of voxels scanned was maximized at each frequency: 5 voxels at 3700 Hz, 10 at 1850 Hz, 20 at 925 Hz, 40 at 462 Hz, and 80 at 231 Hz. For ASAP1 and ASAP2s, respectively, sample sizes at each frequency are as follows: 231 Hz, n = 8 and 9 neurons; 462 Hz, n = 7 and 9, 925 Hz, n = 8 and 22; 1850 Hz, n = 7 and 9; 3700 Hz, n = 6 and 14. All comparisons are between ASAP1 and ASAP2s. ***p<0.001 (t-test with Holm-Bonferroni correction for multiple comparisons). Symbols indicate the mean and error bars show the SEM. (**B**) Noise, defined as the standard deviation of baseline fluorescence, as a function of the scanning frequency in single trials. Sample sizes are the same as in panel A. All comparisons are between ASAP1 and ASAP2s. No significant difference observed between ASAP2s and ASAP1 (Mann-Whitney U-test with Holm-Bonferroni correction for multiple comparisons). Symbols indicate the mean and error bars show the SEM. (**C**) The signal-to-noise ratio (SNR) of single-trial optical responses to current-triggered APs was higher for ASAP2s than ASAP1 at all scanning frequencies. The SNR was calculated as the peak amplitude of optical transients divided by the standard deviation of baseline fluorescence. Sample sizes are the same as in panel A. *p<0.05, ***p<0.001 (t-test with Holm-Bonferroni correction for multiple comparisons). Symbols indicate the mean and error bars show the SEM.

The following figure supplement is available for figure 7:

**Figure supplement 1.** ASAP2f and ASAP1 exhibit a similar dependence on scanning frequency for AP detection.

As expected, averaging over more trials or voxels decreased noise (*Figure 9D,E*). To increase the SNR while imaging single voxels in single trials, we considered that lower acquisition frequencies increase SNR when imaging using ASAP2s (*Figure 7C*). We therefore binned four adjacent time points, effectively increasing the laser dwell time to 200 µs per time point and more than doubling the SNR (*Figure 9F*). With these parameters, we could observe 20 voxels at 231 Hz, with further gains achievable at the same acquisition frequency by increasing dwell time and consequently reducing the number of imaged voxels.

When selecting single voxels for multi-neuron recordings, choosing the brightest membrane voxels would be a useful strategy for maximizing SNR and spike detectability (d'). We evaluated the potential of this strategy by analyzing the d' value for the brightest voxels in our recordings. With ASAP2s, the d' for detecting spikes using single voxels was 17.5 ± 1.2 (n = 23 voxels from 23 neurons), indicating a greater than 99% detection rate and a false-positive rate per frame of less than $10^{-8}$ (*Wilt et al., 2013*). In contrast, the d' for ASAP1 was much smaller at 4.1 ± 0.9 (n = 15 voxels from 15 neurons; p < 0.0001, Mann-Whitney U test), indicating a detection rate of ~94% and a false-positive rate per frame of 0.005. Taken together, these results show that FEVIR using ASAP2s can reliably report APs in single trials, even down to single voxels.

## Tracking spike propagation in neurons with ASAP2s and FEVIR in organotypic slice cultures

The high sensitivity of ASAP2s in single-voxel imaging motivates the deployment of this indicator for monitoring voltage with subcellular resolution with FEVIR. We therefore tested the utility of ASAP2s for optically measuring the speed and attenuation of back-propagating APs in dendrites in organotypic hippocampal slice cultures. Voltage signals were recorded from 25 locations on the dendritic tree at a sampling frequency of 686 Hz and averaged over 31 trials (*Figure 10A*). We observed that

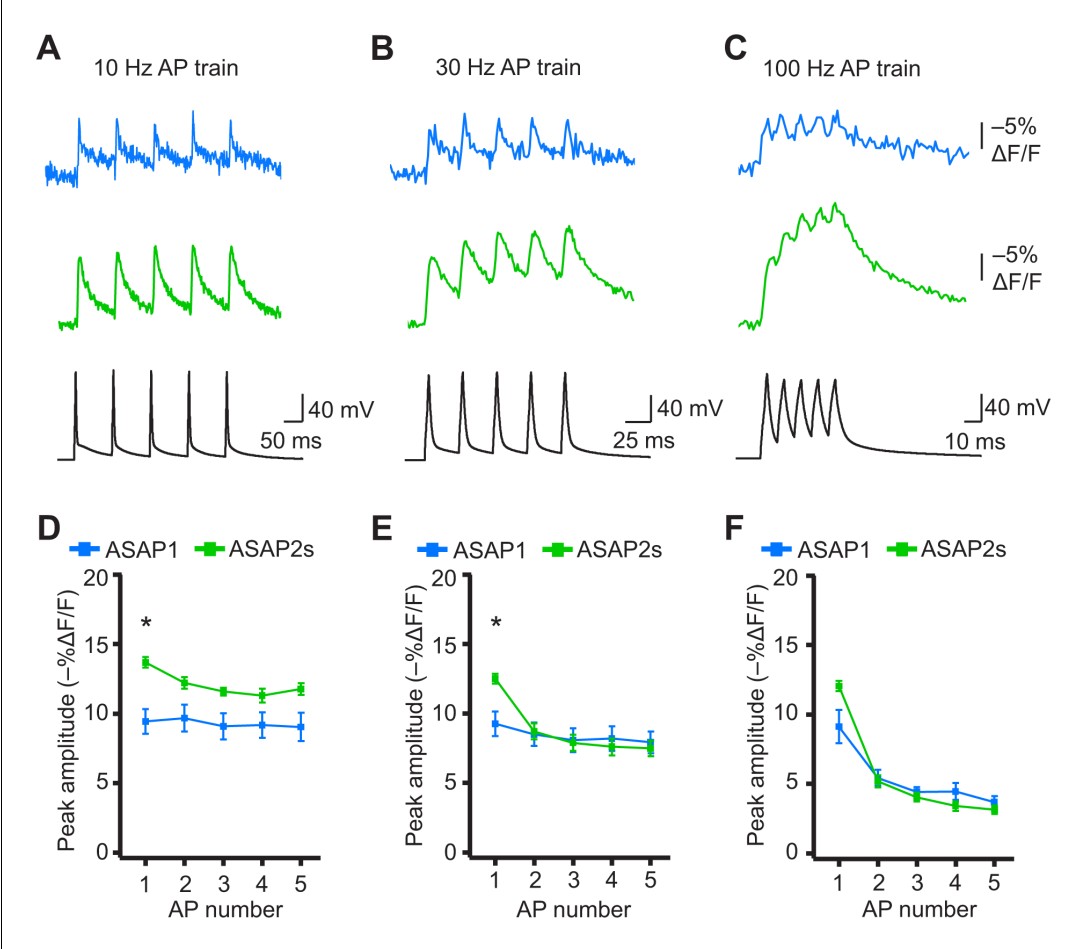

**Figure 8.** Detecting individual spikes in trains of action potentials. (**A–C**) Representative responses to trains of five APs were evoked by current injection at 10 Hz (**A**), 30 Hz (**B**), or 100 Hz (**C**) in organotypic hippocampal slice cultures. Optical recordings were acquired at 925 Hz with 20 voxels per neuron. Traces are the average of 10 trials and were corrected for photobleaching. Blue, ASAP1; green, ASAP2s. (**D–F**) Peak amplitude of ASAP1 and ASAP2s responses to each spike of a train of five APs evoked at 10 Hz (**D**), 30 Hz (**E**), and 100 Hz (**F**) with n = 7 neurons per GEVI. *p<0.05 (t-test corrected with Holm-Bonferroni method for multiple comparisons). Comparisons are between ASAP1 and ASAP2s. Symbols indicate the mean and error bars show the SEM.

The following figure supplements are available for figure 8:

**Figure supplement 1.** Single-trial spike detectability in trains of action potentials.

**Figure supplement 2.** Detecting individual spikes in trains of action potentials using ASAP2f.

the peak amplitude of AP-evoked optical signals gradually decreased with increasing distance from the soma (*Figure 10B,C*, *Figure 10—figure supplement 1A*). For example, response amplitudes were 37.6 ± 7.3% lower at 144.3 ± 3.5 μm from the soma (p=0.0066, n = 4 neurons). This reduction could not be attributed to variation in the levels of resting fluorescence, as we did not detect a correlation between response magnitude and resting fluorescence (*Figure 10—figure supplement 1B*). The degree of attenuation we observed is similar to the 25–35% decreases in AP amplitude previously measured using purely electrophysiological approaches in CA1 pyramidal neurons (*Spruston et al., 1995*; *Golding et al., 2005*) and in cortical neurons (*Stuart et al., 1997*).

Finally, to evaluate whether conduction velocities and latencies of backpropagating APs could be resolved, we scanned two soma voxels and three dendrite voxels on a neuron at a sampling rate of 3700 Hz. Ultrafast sampling allowed us to resolve the kinetics of AP-evoked optical signals

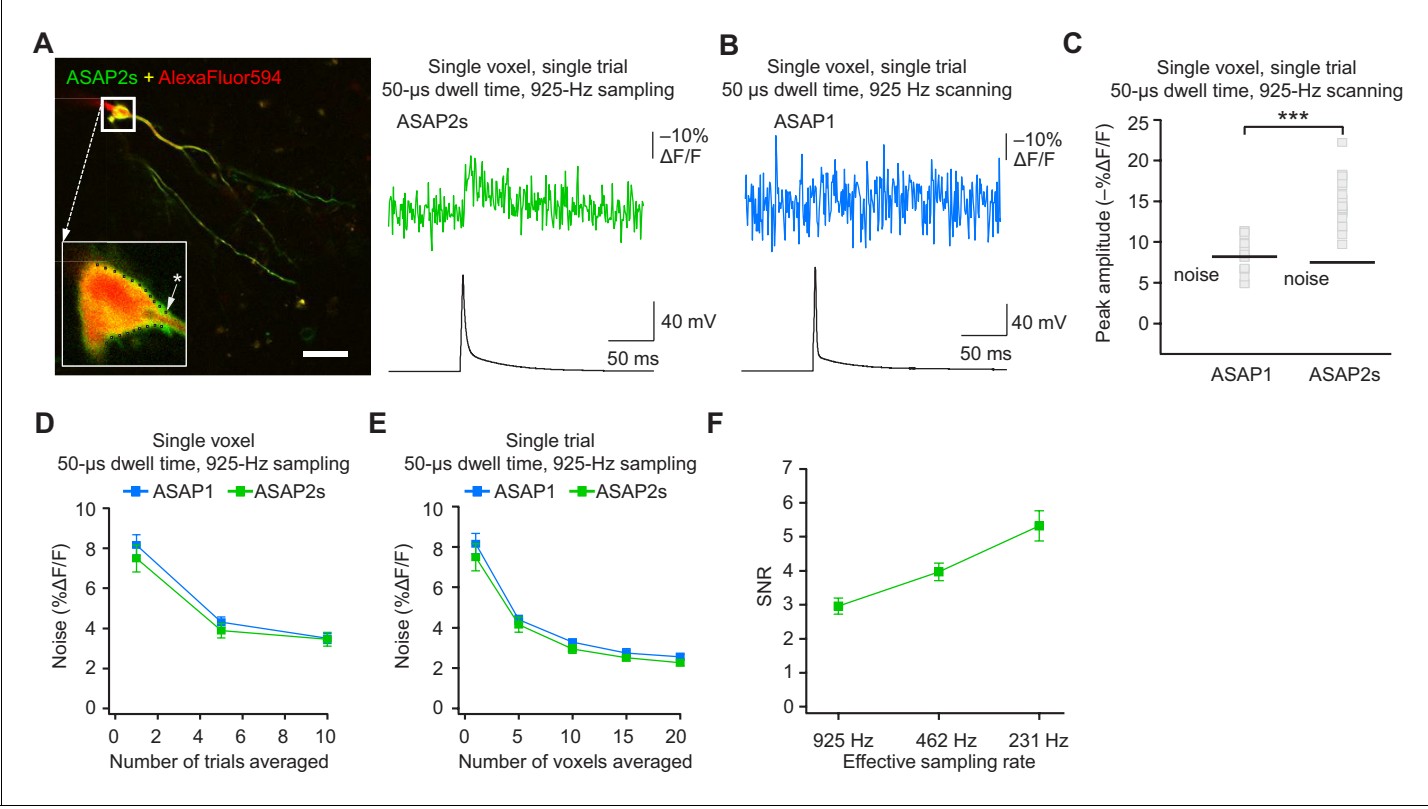

**Figure 9.** Detecting action potentials in single voxels and single trials. (A) Left, overlay image of ASAP2s and AlexaFluor594 fluorescence from a representative neuron in an organotypic hippocampal slice culture. The recording site for this example is shown with an asterisk in the inset. Right, example of a single-trial, single-voxel ASAP2s response to a single current-evoked AP. (B) Example of single-trial, single-voxel ASAP1 response to a single current-evoked AP. (C) Single-trial, single-voxel peak amplitude of ASAP1 and ASAP2s responses compared with the noise level. For each neuron, the peak amplitude was independently measured for all 20 imaged voxels; the mean peak response amplitude for each neuron is shown as a gray square. Black bars correspond to the mean noise level over all cells. n = 15 (ASAP1) and 23 (ASAP2s) neurons. ***p<0.001 (t-test). (D) Single-voxel noise as a function of the number of trials averaged. Symbols indicate the mean and error bars show the SEM. No significant difference was observed between ASAP1 and ASAP2s. n = 15 (ASAP1) and 23 (ASAP2s) neurons. p>0.05 (Mann-Whitney U-test with Holm-Bonferroni correction for multiple comparisons). (E) Single-trial noise as a function of the number of voxels averaged. Symbols indicate the mean and error bars show the SEM. No significant difference was observed between ASAP1 and ASAP2s. n = 15 (ASAP1) and 23 (ASAP2s) neurons. p>0.05 (Mann-Whitney U-test with Holm-Bonferroni correction for multiple comparisons). (F) Resampling ASAP2s responses by binning adjacent timepoints increases the SNR for single-trial single-voxel AP detection. Original timepoints were sampled at 925 Hz with dwell times of 50 μs. Only one voxel was analyzed per neuron, resulting in a slight difference in mean SNR compared with the value derived from the data in panel C and reported in the main text. n = 15 (ASAP1) and 23 (ASAP2s) neurons. For panels D-F, symbols indicate the mean and error bars show the SEM.

The following figure supplement is available for figure 9:

**Figure supplement 1.** Detecting action potentials in single voxels and single trials using ASAP2f.

(**Figure 10D**). For instance, in the example presented, the AP peak was delayed by 1.1 ms in a dendrite at a distance of ~150 μm from the soma (**Figure 10E**). We calculated the mean conduction velocity as 0.16 ± 0.03 m/s (n = 7 neurons, **Figure 10—figure supplement 1C**), similar to a reported velocity of 0.24 m/s for backpropagating APs in hippocampal neurons using electrophysiological methods (**Spruston et al., 1995**). Therefore, the peak amplitude and kinetics of backpropagating APs can be measured across subcellular locations using ASAP2s. We also observed that APs in dendrites were broader than APs in the soma (**Figure 10D**), as previously reported (**Kim et al., 2012**). These results demonstrate that ASAP2s-powered FEVIR permits subcellular voltage imaging with a high SNR, enabling monitoring of neuronal activity simultaneously in multiple subcellular compartments at very high temporal resolution.

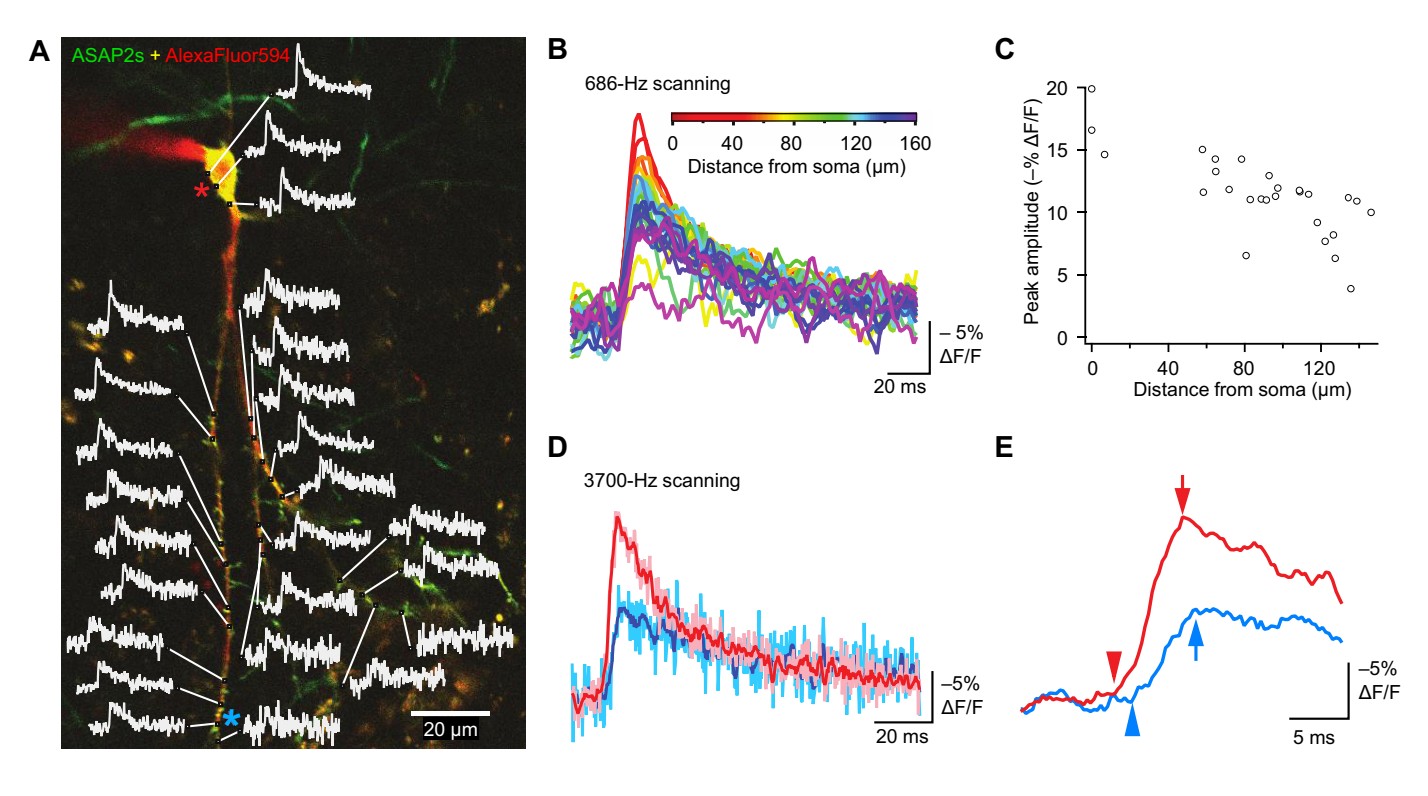

**Figure 10.** Tracking spike propagation in organotypic hippocampal slice cultures with ASAP2s and RAMP imaging. (**A**) Optical responses to a backpropagating, current-evoked AP were recorded at multiple sites in the dendritic arbor. Recorded points at different distances from the cell body are shown with their corresponding traces. Traces are from single voxels imaged at 686 Hz and averaged over 31 trials. This image is an overlay of ASAP2s and AlexaFluor594 fluorescence from a representative neuron. (**B**) Overlay of traces from the points recorded in panel A, color coded to show the distance from the cell body. A five-point moving average was applied to the traces to improve clarity when superimposed. (**C**) Peak response amplitude as a function of distance from the cell body. Each point corresponds to a unique spatial location (voxel) from the neuron shown in panel A. (**D**) Comparison of optical responses to a current-evoked AP recorded at the soma and distal dendrites. The soma trace (light red) is an average of two voxels (near the red star in panel A), while the distal dendrite trace (light blue) is an average of three voxels (near the blue star). Both traces are averages of 23 trials. The darker traces are smoothened traces (five-point moving average). (**E**) The optical traces from panel D were rescaled and aligned to show the delay in initiation time (arrowheads) and peak amplitude (arrows) of the AP in distal dendrites.

The following figure supplement is available for figure 10:

**Figure supplement 1.** Pooled data from optical tracking of spike propagation in multiple neurons using ASAP2s.

## Discussion

In this study, we establish methodology for fast two-photon voltage imaging in vitro, in organotypic slice culture, and in living animals. Our findings have important implications for voltage imaging in intact neuronal preparations. In particular, they establish the suitability of two-photon microscopy and ASAP-family GEVIs, with their fast kinetics and large response amplitudes, for investigations of spontaneous cellular and subcellular voltage changes. Moreover, FEVIR, which combines RAMP imaging and ASAP-family indicators, permits near-simultaneous voltage recording at multiple points subcellular locations. These capabilities are particularly significant given the long-standing quest by neurophysiologists to understand how up to several thousands of synaptic inputs cooperate to trigger an action potential (*Megías et al., 2001*) and to understand the occurrence and functions of dendritic spikes (*Kim et al., 2015*). In general, the ability to record voltage with subcellular precision in single trials should facilitate future interrogation of neuronal physiology by providing easier access to compartments that are either difficult or impossible to access using electrodes, such as small-

caliber dendrites, spines, and boutons. FEVIR could also be used to monitor multiple neurons rather than several locations within the same neuron, enabling studies of circuit function.

Our results demonstrate that the choice of indicator is crucial for high-fidelity monitoring of electrical activity. In *Drosophila* L2 axon terminals, ASAP2s tracked transient voltage responses to changes to visual stimuli with faster kinetics and larger response amplitude than the GEVI ArcLight. In brain slices with FEVIR imaging, the larger response amplitude and slower inactivation kinetics of ASAP2s compared with ASAP1 and ASAP2f enabled improved AP detection, especially at scanning frequencies below 1000 Hz. In particular, ASAP2s allowed AP detection in single voxels in single trials with voxel dwell times of only 50 µs. These parameters enable voltage imaging at ~20 two-photon-addressable points per millisecond, which can be distributed among multiple neurons if desired. Overall, in experimental settings where sensitive detection of spikes is desirable, ASAP2s would generally be preferable to ASAP1 and ASAP2f. However, in specific situations where it is important to track the shape of spikes or other fast voltage transients more accurately, ASAP1 and ASAP2f retain an advantage over ASAP2s due to their faster kinetics.

Opsin-based GEVIs have been used to report transmembrane voltage changes under one-photon illumination, but appear less suitable for two-photon applications. Some single-domain opsins can produce larger responses to APs than ASAP2s in one-photon microscopy (*Hochbaum et al., 2014*), but their responses are greatly attenuated under two-photon microscopy to levels far below that of the ASAP indicators (*Brinks et al., 2015*). Fusions of fluorescent protein domains and voltage-sensitive opsins have led to a new class of GEVIs called FRET-opsin (*Gong et al., 2014*) or electrochromic FRET (*Zou et al., 2014*) indicators. These GEVIs exhibit voltage-induced changes in fluorescence output due to changes in the efficiency of fluorescence resonance energy transfer (FRET) from the fluorescent protein to the rhodopsin under one-photon microscopy (*Gong et al., 2014*; *Zou et al., 2014*). However, these indicators have also shown large reductions in their fluorescence responses under two-photon illumination, with one indicator of this class showing small responses in a previous study (*Brinks et al., 2015*) and two others showing small or undetectable responses in our experiments.

The comprehensive benchmarking of the ASAP indicators across multiple performance metrics and contexts should help biologists determine whether these indicators can enable their experiments. However, we also want to emphasize that experimental preparation, temperature, and illumination condition can impact indicator performance. Indicator expression levels and host cell properties can also alter the impact of GEVIs on plasma membrane capacitance (*Cao et al., 2013*; *Akemann et al., 2009*). Re-evaluation of critical performance metrics and membrane electrical properties is therefore recommended when deploying any indicator in a new experimental context.

While our results illustrate the abilities of fast two-photon imaging of ASAP-family voltage indicators, additional improvements in sensor performance and imaging technology would be useful. For example, further improvements in the brightness or response amplitude of the ASAP indicators would enhance the SNR of responses, improving the reliability of voltage imaging in the brain, especially in challenging imaging situations such as single-trial, single-voxel imaging. Moreover, increasing photostability is paramount for long-term imaging, requiring the development of more photostable probes or methodologies to rapidly shift illumination to non-bleached voxels. In cases where subcellular resolution is not needed, the point spread function of the illumination beam can be expanded to match the size of an entire neuronal cell body (*Prevedel et al., 2016*). This approach would require lower light power to produce identical fluorescence, thereby reducing photobleaching kinetics. Finally, improvements in imaging modalities could increase the number of neurons that can be simultaneously monitored with millisecond-level time resolution.

In summary, we have demonstrated that the combination of two-photon microscopy and ASAP-family voltage indicators can be used to track voltage dynamics with subcellular resolution and millisecond-level temporal precision in brain tissue. Using RAMP microscopy, we have provided the first demonstration that GEVIs can report APs, subthreshold depolarizations, and hyperpolarizations in organotypic slice culture in single trials under two-photon illumination. In addition, we have shown that FEVIR can enable tracking of voltage dynamics across multiple locations of a single neuron. We anticipate that the combination of rapid-scanning two-photon microscopy and ASAP indicators described here will facilitate current and future efforts to understand how neural circuits represent, integrate, and transform information.

## Materials and methods

### Reagent distribution

Expression plasmids and complete sequences for ASAP1 and ASAP2s can be obtained via Addgene (addgene.org). pcDNA3.1/Puro-CAG-ASAP1 is Addgene plasmid #52519 and pcDNA3.1/Puro-CAG-ASAP2s is Addgene plasmid #101274. The sequence for ASAP2s was also deposited to GenBank (accession number MF682491). ASAP transgenic flies are available from the Bloomington *Drosophila* Stock Center (flystocks.bio.indiana.edu).

### Plasmid construction

Plasmids were constructed by standard molecular biology methods and verified by sequencing of all cloned fragments.

### In vitro experiments with cell lines, stem-cell-derived cardiomyocytes, and neuronal cultures

The ASAP variants and ArcLight Q239 were all expressed from pcDNA3.1/Puro-CAG (*Lam et al., 2012*), a plasmid vector with the strong synthetic promoter CAG. As previously reported for ASAP1 and ArcLight Q239 (*St-Pierre et al., 2014*), we subcloned all indicators between the NheI and HindIII sites of this plasmid. All indicators had identical Kozak sequences. For experiments to compare indicator brightness, the red fluorescent protein FusionRed (*Shemiakina et al., 2012*) was fused to the C-terminus of the ASAP variants. To ensure optimal folding of both ASAP indicators and FusionRed, we separated these two domains with a flexible glycine/serine linker (GSGGSGGSG). ASAP1::EGFP, the reference indicator for photostability experiments, was constructed by replacing the fluorophore of ASAP1 with EGFP (V2 to K239). For experiments comparing two-photon excitation spectra, we expressed EGFP using the ubiquitous EF-1α promoter cloned in a pLenti (also called pLECYT) plasmid (*Boyden et al., 2005*).

### Experiments in *Drosophila*

ASAP1, ASAP2s, and MacQ-Citrine indicators were cloned downstream of the Upstream Activating Sequence (UAS) of the pJFRC7-20XUAS vector (*Pfeiffer et al., 2010*).

### Experiments in organotypic slice cultures

Ace2N-4AA-mNeon was amplified from a template generously provided by M. Schnitzer and subcloned into pcDNA3.1/Puro-CAG using methods described above. All slice culture experiments used pcDNA3.1/Puro-CAG expression plasmids for expressing the ASAP indicators and Ace2N-4AA-mNeon.

### HEK293 — cell culture

HEK293A (Thermo Fisher Scientific, Waltham, MA; RRID:CVCL_6910) and HEK293-Kir2.1 (*Zhang et al., 2009*) cell lines were confirmed to be free of mycoplasma contamination using the MycoAlert Mycoplasma Detection Kit (Lonza, Switzerland). HEK293-Kir2.1 cells were confirmed to have a polarized resting membrane potential ($-77 \pm 1.2$ mV, mean $\pm$ sem, n = 10 cells) consistent with the original report of this cell line (*Zhang et al., 2009*). Given that cell lines were used as expression systems for characterizing voltage indicators rather than for biological discovery and that the performance metrics of voltage indicators in these cells are consistent with published results and with further experiments in neurons, cell lines were not authenticated using DNA profiling analysis. Cells were maintained in high-glucose Dulbecco's Modified Eagle Medium (DMEM, GE Healthcare, Chicago, IL) supplemented with 5% fetal bovine serum (FBS, Thermo Fisher Scientific) and 2 mM glutamine (Sigma-Aldrich, St.Louis, MO) at 37°C in air with 5% $CO_2$. Cells were plated onto glass-bottom 24-well plates (In vitro Scientific) for standard imaging or onto uncoated no. 0 12 mm coverslips (Glaswarenfabrik Karl Hecht GmbH, Germany) for patch clamp experiments. Transfections were carried out using FuGene HD (Promega, Madison, WI) according to the manufacturer's instructions, except that cells were transfected at ~50% confluence with lower amounts of DNA (200 ng) and transfection reagent (0.6 μL) to reduce cell toxicity.

## HEK293 — patch clamping and voltage imaging

At 2 days post transfection, cells were patch-clamped at 22°C using borosilicate glass electrodes with resistances of 3.5 to 5.0 MOhm attached to an Multiclamp 700B amplifier (Molecular Devices, Sunnyvale, CA). Cells were superfused in a chamber mounted on the stage of an Axiovert 100M inverted microscope with a 40×/1.3-numerical aperture (NA) oil-immersion objective (Zeiss, Germany). The extracellular solution contained 110 mM NaCl, 26 mM sucrose, 23 mM glucose, 5 mM HEPES-Na, 5 mM KCl, 2.5 mM $CaCl_2$, and 1.3 mM $MgSO_4$, adjusted to pH 7.4. The intracellular (pipette) solution contained 115 mM potassium gluconate, 10 mM HEPES-Na, 10 mM EGTA, 10 mM glucose, 8 mM KCl, 5 mM $MgCl_2$ and 1 mM $CaCl_2$, adjusted to pH 7.4.

Cells were illuminated with a high-power light-emitting diode (LED, UHP-MIC-LED-460, Pryzmatix, Israel) through a 480/20 nm filter at a power density of 4 to 24 mW/mm$^2$ at the sample plane. Emitted photons were filtered using a 525/50 nm filter. Images were acquired using an iXon 860 electron multiplying charge coupled device camera (Andor, UK) cooled to −80°C and set to Frame Transfer mode. Fluorescence traces were corrected for photobleaching. Voltage and fluorescence traces were analyzed using custom scripts written in MATLAB (Mathworks, Natick, MA). Voltage traces were corrected for the junction potential post hoc.

Fluorescence traces were acquired while cells were voltage clamped in whole-cell mode. Unless otherwise indicated, step voltage depolarizations were applied to change the membrane potential from a holding voltage of −70 mV to voltages ranging from −120 mV to 50 mV for 1.0 s. For these voltage step experiments, we captured images at 200 Hz without binning. While fluorescence responses were measured from pixels at the perimeter of the cell (the plasma membrane), values obtained over the entire cell are nearly identical given the excellent membrane localization of ASAP1 and ASAP2s. For experiments with individual or trains of artificial AP waveforms, we captured images at 1 kHz with 4 × 4 binning. The AP waveform, derived from a recording of a hippocampal neuron AP, has a full width at half maximum of 4.0 ms and peak amplitude of 100 mV. The fluorescence response was measured as described above. For experiments to determine sensor response kinetics, we increased the sampling frame rate to 2.5 kHz by cropping the imaged area down to 64 × 64 pixels and increasing binning to 8 × 8 pixels. The resulting image thus contained 64 pixels (8 × 8); fluorescence was calculated by summing all 64 pixels. Command voltage steps were applied for 1 s; three identical voltage steps were measured for every cell. Models of the form $a \bullet e^{-bt} + c \bullet e^{-dt}$ were applied to the rising and falling portions of the mean fluorescence trace using MATLAB (Mathworks).

## HEK293 — two-photon excitation spectra

HEK293-Kir2.1 cells were transfected as described above. Cells were imaged with an A1R MP + microscope (Nikon, Japan) fitted with 20× 0.75NA dry objective, a 525/50 nm filter, gallium arsenide phosphide (GaAsP) detectors, and a titanium:sapphire Chameleon Ultra I laser (Coherent, Santa Clara, CA) with a 80 MHz repetition rate and a pulse width of 140 fs (measured at 800 nm). The laser was tuned between 700 nm and 1040 nm, keeping the laser power at 10 mW across the spectrum. However, this system did not pre-compensate laser pulses for dispersion in the microscope optical path. Pulse width is therefore expected to vary with wavelength (*Müller et al., 1998*), thereby impacting two-photon excitation absorption efficiency and distorting excitation spectra over those acquired by a pre-compensated system. Laser scanning was performed using galvanometric mirrors. Each image pixel was sampled with a dwell time of 12.1 μs. To evaluate the impact of photobleaching, we acquired images at 900 nm both before and after each wavelength scan. We also compared spectra obtained by scanning from 700 nm to 1040 nm to those produced in the reverse direction (1040 nm to 700 nm). Both methods demonstrated that photobleaching was negligible, and we therefore did not correct for photobleaching.

## HEK293 — quantifying GEVI brightness

For quantifying the brightness of ASAP variants, we used a HEK293 cell line expressing the inwardly rectifying Kir2.1 channel (HEK293-Kir2.1, [*Zhang et al., 2009*]). This cell line, a generous gift of Gui-Rong Li, has a resting membrane potential of approximately –77 mV, similar to that of primary hippocampal neurons. Cells were transfected with pcDNA3.1/puro-CAG plasmids expressing ASAP1-FusionRed or ASAP2s-FusionRed (see: Plasmid construction). As discussed previously, FusionRed

served to normalize for cell-to-cell differences in expression level, thus allowing brightness to be quantified as the ratio of green fluorescence (from the ASAP indicators) to red fluorescence (from the FusionRed standard). Two days post-transfection, cells were superfused with the same extracellular solution we used for electrophysiological recordings in HEK293A cells and imaged with an inverted A1R MP + microscope (Nikon) fitted with a 40× 1.3-NA oil immersion objective.

To evaluate brightness under one-photon illumination, we used the SpectraX light engine (Lumencor). GFP was excited with cyan light filtered with a 470/24 nm excitation filter and at a power density of 87 mW/mm$^2$. FusionRed was excited with yellow light filtered with a 555/15 nm excitation filter and at a power density of 151 mW/mm$^2$. Emitted photons were filtered with 520/23 nm (GFP) and 597/39 nm (FusionRed) filters and acquired with an ORCA Flash4.0 V2 C11440-22CU (Hamamatsu, Japan) scientific CMOS camera set to 4 × 4 pixels binning and cooled to −10°C.

To evaluate brightness under two-photon illumination, we used a titanium:sapphire Chameleon Ultra I laser (Coherent) with a 80 MHz repetition rate and a pulse width of 140 fs (measured at 800 nm). The laser was tuned to 900 nm (ASAP) or 1040 nm (FusionRed). Laser pulses were not pre-compensated for dispersion in the microscope optical path. We excited ASAP variants at 900 nm rather than at its peak (920 nm) to match the excitation wavelength used in our organotypic slice culture experiments, which we performed using a random-access multi-photon (RAMP) system that is not compatible with wavelengths exceeding 900 nm. Power was adjusted to 30 mW (ASAP indicators) and 12 mW (FusionRed). This system did not pre-compensate laser pulses for dispersive effects of the microscope optical path. Laser dwell time was set to 12.1 μs, and scanning was performed using galvanometric mirrors. Emitted light was filtered using 525/50 nm (ASAP) or 605/70 (FusionRed) filters. Images were acquired with gallium arsenide phosphide (GaAsP) detectors.

## HEK293 — quantifying GEVI photostability

For quantifying the photostability of ASAP variants, we transfected HEK293-Kir2.1 cells as described above with the corresponding pcDNA3.1/Puro-CAG expression plasmids (see: Plasmid construction). At 2 days post-transfection, cells were superfused with the same extracellular solution we used for electrophysiological recordings in HEK293A cells.

To obtain the one-photon photostability data of *Figure 1—figure supplement 3A–B*, cells were imaged using an Axiovert 100M microscope (Zeiss) fitted with a 40× 1.3-NA oil-immersion objective. Cells were continuously illuminated with a high-power light-emitting diode (LED, UHP-MIC-LED-460, Pryzmatix) through a 480/20 nm filter at a power density of 11 mW/mm$^2$. Emitted photons were filtered using a 525/50 nm filter, and images were acquired with an ORCA Flash4.0 V2 C11440-22CU (Hamamatsu) scientific CMOS camera set to 4 × 4 pixels binning and cooled to −10°C. Due to relocation of one of the authors, data for *Figure 1—figure supplement 3C–F* was acquired with a different system. For these figures, cells were imaged with an Eclipse Ti-E microscope (Nikon) fitted with a 40× 1.3-NA oil immersion objective. Cells were illuminated with a SpectraX solid-state light source (Lumencor, Beaverton, OR) through a 470/24 nm filter and at a power density of 87 mW/mm$^2$. Emitted photons were filtered using a 520/23 nm filter, and acquired with an ORCA Flash4.0 V2 C11440-22CU (Hamamatsu) scientific CMOS camera set to 4 × 4 pixels binning and cooled to −10°C.

To obtain the two-photon photostability data presented in *Figure 1—figure supplement 4A–B*, cells were imaged with an Ultima Multiphoton Microscopy System (Bruker, Billerica, CA) equipped with a Mai Tai HP Deep See Ti:sapphire laser with <80 fs pulses at 800 nm (Spectra-Physics, Santa Clara, CA), galvanometric mirrors for laser scanning, a 60 × 0.9 NA objective, a 525/50 nm emission filter, and non-descanned multi-alkali photomultiplier tubes (Hamamatsu). The laser was tuned to 920 nm and laser pulses were pre-compensated for dispersion in the microscope optical path. Laser power, laser dwell time, and scanning frequency are specified in the corresponding figure legend. Due to relocation of one of the authors, data for all other two-photon photostability experiments were acquired with different systems. For *Figure 1—figure supplement 4E* cells were imaged with an A1R MP+ microscope (Nikon) fitted with a titanium:sapphire Chameleon Ultra I laser (Coherent) with 80 MHz repetition rate and pulse width of 140 fs (measured at 800 nm), galvanometric mirrors for laser scanning, a 40 × 1.3 NA oil immersion objective, a 525/50 nm emission filter and gallium arsenide phosphide (GaAsP) detectors. The laser was tuned to 900 nm and laser pulses were not pre-compensated for dispersion in the microscope optical path. Laser power, laser dwell time, and scanning frequency are specified in the corresponding figure legend. For *Figure 1—figure*

*supplement 1–C,F,G*, cells were imaged on an LSM 7 MP microscope (Zeiss) fitted with a titanium: sapphire Chameleon Ultra II laser (Coherent) with 80 MHz repetition rate and pulse width of 140 fs (measured at 800 nm), a 20×/1.0-NA objective, mirror galvanometers for laser scanning, and gallium arsenide phosphide (GaAsP) detectors. The laser was tuned to 900 nm and laser pulses were not pre-compensated for dispersion in the microscope optical path. Since this equipment was only used to image single-labeled specimens (GFP only), we improved detection sensitivity by removing the emission filter. Laser power, laser dwell time, and scanning frequency are specified in the corresponding figure legends.

For both one-photon and two-photon photobleaching time series, fluorescence from the entire cell was used to compute optical traces. To quantify the photobleaching time constants, we fit the fluorescence traces with single- and multi-exponential fits using MATLAB (Mathworks).

## Stem-cell-derived cardiomyocytes — generation and cell culture

All stem cell protocols were approved by the Stanford University Human Subjects Research Institutional Review Board (IRB). Cultures were maintained in a 5% $CO_2$/air environment. We used authenticated H9 human embryonic stem cells from the WiCell Research Institute. Human-induced pluripotent stem cells (iPSCs) were generated from skin fibroblasts as described previously (*Ebert et al., 2014*). iPSCs were authenticated by karyotyping to confirm genomic integrity. Routine analyses of pluripotency were conducted by immunostaining pluripotency markers. Human embryonic stem cells (hESCs) and iPSCs were confirmed to be free of mycoplasma contamination using the MycoAlert Mycoplasma Detection Kit (Lonza).

hESCs and iPSCs were maintained on Matrigel-coated plates (BD Biosciences, San Jose, CA) in Essential 8 Medium (Gibco, Thermo Fisher Scientific) and were differentiated into cardiomyocytes as described (*Lian et al., 2012*). Briefly, stem cells were seeded in 6-well plates pre-coated with Matrigel. After reaching 80% confluence, cells were treated for 48 hr with 6 µM CHIR99021 (Selleckchem. com, Houston, TX) in RPMI 1640 Medium (Gibco, Thermo Fisher Scientific) with B-27 serum-free insulin-free supplement (Gibco, Thermo Fisher Scientific). Cells were then transferred to the same medium without CHIR99021 for 24 hr and then treated with 5 µM IWR-1 (Sigma-Aldrich) for 2 days. The media was replaced with fresh media without IWR-1 for another 2 days, before finally switching to RPMI 1640 medium with insulin-containing B-27 supplement. Beating cells were observed at 9 to 11 days post-differentiation and replated to improve attachment and to adjust density. Cardiomyocytes were purified using glucose-free RPMI + B27 medium for two-three rounds, each round lasting two days. Between each round, cells were allowed to recover in normal RPMI + B27 medium containing glucose for two days. The resulting cultures were typically more than 90% pure as determined by the percentage of TNNT2-positive cells by flow cytometry.

## Stem-cell-derived cardiomyocytes — voltage imaging

To prepare cells for imaging, hESC- or iPSC-derived cardiomyocytes were dissociated into single cells with Accutase (Thermo Fisher Scientific) for 20 min. Cells were re-seeded at a density of 50,000 cells/cm$^2$ on 12 mm round borosilicate coverglass (Carolina Biological Supply Company, Burlington, CA) coated with Matrigel (BD Biosciences) placed within individual wells of 24-well plates. Cells were recovered in RPMI medium supplemented with B27 plus insulin (Thermo Fisher Scientific) for 3 to 4 days to allow them to restart beating. Media was replaced every 1 to 2 days.

At 24 to 30 days post-differentiation, cardiomyocytes were transfected with 2.5 µL Lipofectamine 2000 (Thermo Fisher Scientific) and 500 ng of sensor DNA as described in the manufacturer's instructions. At 2 days post-transfection, cells were superfused with the same extracellular solution used for electrophysiological recordings in HEK293A cells. Cells were imaged at 100 Hz with an Axiovert 100M inverted microscope (Zeiss) equipped with a light-emitting diode (LED, UHP-MIC-LED-460, Pryzmatix), a 480/20 nm excitation filter, a 525/50 nm emission filter, a 40×/1.3- NA oil-immersion objective. Cells were illuminated at a power density of 11 mW/mm$^2$ at the sample plane. For experiments with hESC-CMs, image acquisition was performed using Solis (Andor) driving a DU-860 EM-CCD (Andor) cooled to −80°C and set to frame transfer mode. For experiments with iPSC-CMs, we used HCImage (Hamamatsu) to acquire images from an ORCA-Flash4.0 V2 C11440-22CU scientific CMOS camera (Hamamatsu) with 4 × 4 pixel binning and cooled to −10°C. Both cameras could be used interchangeably, although when using a CameraLink interface, the Flash4.0 camera enabled

acquisition of a larger field of view (512 × 512 pixels at 26 µm/pixel after 4 × 4 binning) compared to the DU-860 (120 × 120 pixels at 20 µm/pixel). Images were analyzed with custom MATLAB (Mathworks) scripts. Fluorescence from the entire cell was used to compute optical traces. Traces were corrected for photobleaching.

## Neuronal cell cultures — preparation and transfection

Unless otherwise indicated, primary hippocampal or cortical neurons were dissected from Sprague-Dawley rats on embryonic days 21–22 and digested with 0.03% trypsin (Sigma-Aldrich) in Dulbecco's Modified Eagle Media (DMEM, HyClone, GE Healthcare) for 20 min at 37°C in air with 5% $CO_2$. Neurons were then dissociated by gentle trituration in Hanks' Balanced Salt Solution (HBSS, Thermo Fisher Scientific) and washed twice in HBSS. Neurons were plated at 3.5 × 10$^4$ cells cm$^{-2}$ on 12 mm no. 0 coverslips (Glaswarenfabrik Karl Hecht GmbH) within wells of 24-well plates. Prior to plating, each coverslip was pre-coated for 24 hr with >300 kDa poly-D-lysine (Sigma-Aldrich) in PBS and washed three times with distilled water. Neurons were cultured overnight at 37°C in air with 5% $CO_2$ in Neurobasal with 1 × B27 supplement (Thermo Fisher Scientific), 2 mM GlutaMAX (Thermo Fisher Scientific), and 10% FBS. The following day, 90% of the medium was replaced with identical medium without FBS. Cytosine β-d-arabinofuranoside (Sigma-Aldrich) was added to a final concentration of 2 µM when glia reached >70% confluence, typically around 5 days in vitro (DIV). At 7 DIV, 50% of the media was replaced with fresh media without serum. Neurons were transfected at 7 to 9 days DIV using 0.5 to 0.75 µL of Lipofectamine 2000 (Thermo Fisher Scientific) and 800 ng of total DNA per well of a 24-well plate. Given the strong promoter (CAG) driving indicator expression, we diluted indicator expression plasmids with buffer (unexpressed) plasmids. Specifically, each well was transfected with 400 ng of indicator expression plasmid and 400 ng of pNCS, an empty bacterial expression plasmid (*Lam et al., 2012*).

## Neuronal cell cultures — confocal imaging

### Cortical neurons

Images for *Figure 2G* were obtained as follows. Rat cortical neurons were transfected at 8 DIV and imaged 3–4 days post-transfection in HBSS supplemented with 10 mM HEPES pH 7.4, 1 × B27, 2 mM GlutaMAX, and 1 mM sodium pyruvate on an IX81 microscope with a FluoView FV1000 laser-scanning confocal unit. Fluorescence excitation was delivered using a 488 nm argon laser through a 40×/1.3-NA oil-immersion objective (Olympus, Japan). Emission was passed through a 530/40 nm emission filter. Z-sections were imaged using a 1-Airy pinhole setting and two-pass Kalman filtering. A maximum-intensity projection was generated from two to three sections spaced 2 µm apart.

### Hippocampal neurons

Data for *Figure 2—figure supplement 2* were obtained as follows. Rat hippocampal neurons were dissected and cultured as described above but using neurons from rats at embryonic day 18. After coating plates with poly-lysine, we also incubated plates in 20 µg/mL laminin (Sigma-Aldrich) for at least 5 hr. We observed that laminin treatment significantly promoted growth and maturation of these younger neurons, producing neurons with more extensive dendritic arbors. Since glial growth was reduced using E18 compared to E21 neurons, we did not add Cytosine β-d-arabinofuranoside.

Cells were imaged 2 days post-transfection in HBSS supplemented with 10 mM HEPES pH 7.4, 1 × B27, 2 mM GlutaMAX, and 1 mM sodium pyruvate on an LSM 780 confocal microscope (Zeiss), fitted with a 40×/1.4-NA objective, a 488 nm argon laser model, mirror galvanometers for laser scanning, a 530/80 nm emission filter, and gallium arsenide phosphide (GaAsP) detectors. Each image pixel was sampled with a dwell time of 12.1 µs per pixel. Z-sections were imaged using a 1-Airy pinhole setting. Frames correspond to the average of four scans. Maximum-intensity projections were generated from two to three sections spaced 0.4 µm apart.

## Neuronal cell cultures — patch clamping and voltage imaging

At 11 to 13 DIV (2 to 4 days post-transfection), cultured neurons were patch-clamped at 22°C using the same procedures as when patching HEK293A cells (above). Cells were imaged with an Axiovert 100M inverted microscope (Zeiss) equipped with a 40×/1.3- NA oil-immersion objective. Cells were illuminated with a high-power light-emitting diode (UHP-MIC-LED-460, Pryzmatix) through a 480/20

nm filter at a power density of 24 mW/mm² at the sample plane. Emitted photons were filtered using a 525/50 nm filter. Images were captured at 1000 Hz with 4 × 4 binning using an iXon 860 electron multiplying charge coupled device camera (Andor) cooled to −80°C in frame transfer mode. Fluorescence response was measured in all pixels from the cell body. Fluorescence traces were acquired while cells were current-clamped in whole-cell mode. For all experiments, fluorescence traces were corrected for photobleaching. To generate APs, 700 to 1100 pA of current was injected for 1 ms. Analyzed neurons had the following characteristics: an access resistance less than 15 MOhm, a membrane resistance greater than 10 times the access resistance, and APs with peak height >0 mV and width <5 ms at −20 mV. Electrode voltages were recorded using pClamp (Molecular Devices). Voltage and fluorescence traces were analyzed using custom scripts written in MATLAB (Mathworks). Voltage traces were corrected for the junction potential post hoc.

## *Drosophila* — generation and genotypes of transgenic flies

*UAS-ASAP1*, *UAS-ASAP2s,* and *UAS-MacQ-Citrine* (cf. Plasmid Construction) were inserted into the *attP40 phiC31* landing site by injection of fertilized embryos (Rainbow Transgenic Flies, Inc.). *UAS-ASAP2s* was additionally inserted into the *VK00005 phiC31* landing site though all experiments presented here used the *attP40* insertion. ArcLight was expressed using *UAS-ArcLight* inserted into *attP40* (*Cao et al., 2013*), Ace2N-2AA-mNeon was expressed using *UAS-Ace2N-2AA-mNeon* inserted into *attP40* (*Gong et al., 2015*), jRGECO1b was expressed using *UAS-jRGECO1b* inserted into *VK00005* (*Dana et al., 2016*), and GCaMP6f was expressed using *UAS-GCaMP6f* also inserted into *attP40* (*Chen et al., 2013*). The indicators were specifically expressed in L2 cells using the *21D-Gal4* driver (*Rister et al., 2007*). The genotypes of flies imaged were:

w/+; UAS-ASAP1/+; 21D-Gal4/+
w/+; UAS-ASAP2s/+; 21D-Gal4/+
w/+; UAS-ArcLight/+; 21D-Gal4/+
w/+; UAS-MacQ-Citrine/+; 21D-Gal4/+
w/+; UAS-Ace2N-2AA-mNeon/+; 21D-Gal4, UAS-jRGECO1b/+
w/+; UAS-GCaMP6f/+; 21D-Gal4/+

FlyBaseIDs, Research Resource Identifiers (RRIDs) and FlyBase Record URLs for the fly lines are as follows:

| Fly line | FlyBase ID | RRID | FlyBase record URL |
| --- | --- | --- | --- |
| w/+; UAS-ASAP1/+ | 65412 | RRID:BDSC_65412 | http://flybase.org/reports/FBst0065412.html |
| w/+; UAS-ASAP2s/+ | (in process) | (in process) | (in process) |
| w/+; UAS-ArcLight/+ | 51057 | RRID:BDSC_51057 | http://flybase.org/reports/FBst0051057.html |
| w/+; UAS-MacQ-Citrine/+ | n/a | n/a | n/a |
| w/+; UAS-Ace2N-2AA-mNeon/+ | 64317 | RRID:BDSC_64317 | http://flybase.org/reports/FBst0064317.html |
| w/+; UAS-GCaMP6f/+ | 42747 | RRID:BDSC_42747 | http://flybase.org/reports/FBst0042747.html |
| 21D-Gal4 | FBti0128604 | n/a | http://flybase.org/reports/FBti0128604.html |

## *Drosophila* — husbandry

All flies used for imaging, except those expressing MacQ-Citrine or Ace2N-2AA-mNeon, were raised on standard molasses food at 25°C on a 12/12 hr light-dark cycle. Female flies of the appropriate genotypes were collected on CO₂ within 1 day of eclosion and imaged at room temperature (20°C) at 5 to 6 days after eclosion. MacQ-Citrine female flies were raised throughout life at 25°C on standard molasses food supplemented with 100 µM all-trans-retinal (Sigma-Aldrich) to ensure that the MacQ opsin had adequate quantities of retinal cofactor; flies were kept in the dark to prevent retinal degradation. Flies were collected on CO₂ within 1 day of eclosion and imaged at room temperature (20°C) at 5 to 6 days after eclosion. Following a protocol from the original description of Ace2N-2AA-mNeon (*Gong et al., 2015*), Ace2N-2AA-mNeon female flies were raised to adulthood at 25°C on standard molasses food and collected on CO₂ within 1–3 days of eclosion. At this time, they were transferred to standard molasses food supplemented with 400 µM all-trans-retinal (Sigma-Aldrich) to

ensure that the Ace2N opsin had adequate quantities of retinal cofactor; flies were kept in the dark to prevent retinal degradation. Flies were imaged after 6 days on retinal food.

## Drosophila — surgeries, stimulus presentation, and two-photon imaging

Flies for imaging were cold anaesthetized, positioned in a fly-shaped hole cut in steel foil such that their heads were tilted forward approximately 60° to expose the back of the head capsule above the foil while leaving most of the retina below the foil and then affixed in place with UV-cured glue (NOA 68T from Norland Products Inc., South Brunswick Township, NJ). The brain was exposed by removing the overlying cuticle and fat bodies with fine forceps, and an oxygenated saline-sugar solution (*Wilson et al., 2004*) was flowed over the fly.

L2 axon terminals in the medulla were imaged with a TCS SP5 II two-photon microscope (Leica) with an HCX APO 20×/1.0-NA water immersion objective (Leica) and a titanium:sapphire Chameleon Vision II laser (Coherent) with 80 MHz repetition rate and pulse width of 140 fs (measured at 800 nm). The laser pulses were pre-compensated for dispersion in the microscope optical path. Unless otherwise indicated, the excitation wavelength was 920 nm and 5 to 15 mW of power was applied at the sample. Emitted photons were filtered with a 525/50 nm filter, except for jRGECO1b, for which emitted photons were filtered with a 585/40 nm filter.

Visual stimuli were generated with custom-written scripts using C ++ and OpenGL and presented using a digital light projector as described previously (*Clark et al., 2011*). The visual stimulus was projected onto a coherent fiber optic bundle that then re-projected onto a rear-projection screen positioned approximately 4 cm anterior to the fly that spanned 80° of the fly's visual field horizontally and 50° vertically. Immediately prior to being projected onto the screen, the stimulus was filtered with a 447/60 nm or a 482/18 nm bandpass filter to prevent its detection by the microscope PMTs. The stimulus was updated at 120 Hz and had a radiance of approximately 30 mW sr$^{-1}$ m$^2$. The imaging and the visual stimulus presentation were synchronized as described previously (*Freifeld et al., 2013*). Following this procedure, the time of stimulus onset relative to the start of imaging varied by up to one stimulus frame (8.33 ms). To compensate for this, the average delay was measured (6.25 ms), and all imaging data was shifted in time by this delay. All data were acquired at a constant frame rate of 38.9 Hz using a frame size of 200 × 20 pixels, a line scan rate of 1400 Hz, and bidirectional scanning. Imaging time per fly never exceeded 1 hr.

## Drosophila – image analysis

Raw images in each time series were aligned in x-y to correct for motion artifacts using a macro based on the plug-in Turboreg in ImageJ (National Institutes of Health, Bethesda, MD); if motion artifacts were too severe to be corrected, the entire time series was not analyzed further. The remaining analysis was performed with MATLAB (Mathworks). Regions of interest (ROIs) around individual L2 medulla terminals were manually selected in the time-series-averaged image. Intensity values for the pixels within each ROI were averaged and the mean background value was subtracted. To correct for bleaching, the time series for each ROI was fit with the sum of two exponentials, and in the calculation of $\Delta F/F = (F(t) – F_0)/ F_0$, the fitted value at each time t was used as $F_0$. The stimulus-locked average was computed for each ROI by reassigning the timing of each imaging frame to be relative to the stimulus transitions (dark to light or light to dark) and then computing a simple moving average with a 25 ms averaging window and a shift of 8.33 ms (120 Hz). As the screen on which the stimulus was presented did not span the fly's entire visual field, some imaged ROIs corresponded to cells with receptive fields outside of the area covered by the stimulus (empirically,~30% of the ROIs imaged per fly); the ~70% of responding ROIs were identified as those whose responses at time-matched points during the dark flash and during the light flash were significantly different (t-test, p<0.01) for at least three consecutive time points. For MacQ-mCitrine, none of the ROIs met this criterion; as such, all imaged ROIs were presented in *Figure 3B*. For flies expressing each of the other indicators, there were approximately the same fraction of responding ROIs in each fly imaged. Because of the low amplitude of the Ace2N-2AA-mNeon responses, the responses of the co-expressed jRGECO1b calcium indicator were used to identify responding ROIs. For the upper traces in *Figure 3B*, we first determined the mean response of each ROI across all trials and then calculated the mean response across all responding ROIs. The lower traces in *Figure 3B* are the moving-

average response of single example ROIs (that are approximately in the 75th percentile for response amplitude for all ROIs of that indicator; colored traces), and 0.6 s single-trial excerpts of the $\Delta F/F$ trace for the same ROIs presented in the single cell examples (plotted at the imaging frame rate of 38.9 Hz; gray traces). Quantification metrics were calculated on each ROI's mean response. The peak response ($A_{max}$) during each phase (depolarization and hyperpolarization) was the largest value of $|\Delta F/F|$. The time to peak ($t_{peak}$) was the time at which this peak response occurred, relative to the start of the stimulus. The decay of the response from the peak was fit with a single exponential and the time constant of this fit was $\tau_{decay}$. All responding cells (ROIs) were included in the comparisons in *Figure 3D*, except for $\tau_{decay}$, for which we only included cells whose exponential decay could be fit with r-squared values greater than 0.5, as ROIs with values smaller than this were those that were too noisy to be fit.

## Organotypic hippocampal slice cultures – preparation and electroporation

Organotypic hippocampal slices were prepared using 400-μm thick transverse hippocampal slices cut from 6- to 8-day-old male Wistar rats (*Stoppini et al., 1991*). Animals were anesthetized, the brain was quickly removed, and the resected hippocampus was transferred into cutting solution and cut using a McIlwain tissue chopper (Ted Pella, Redding, CA). Following dissection, three slices were plated on Millicell culture plates (EMD Millipore, Billerica, MA). Slices were kept at a liquid/gas interface in a controlled atmosphere (95% $O_2$, 5% $CO_2$) chamber at 37°C for 7 to 14 days before use. Fresh medium was provided twice a week. On days 2 to 4 following slice preparation, expression plasmids were bulk electroporated with an Grass Technologies electrical stimulator (Natus, Pleasanton, CA). 10 pulses (50 V) were delivered at 10 Hz. Slices were incubated for 7 days following transfection before starting experiments.

## Organotypic hippocampal slice cultures – electrophysiology

During recordings, slices were continuously perfused with artificial cerebrospinal fluid containing 124 mM NaCl, 25 mM NaHCO$_3$, 2.5 mM KCl, 1.2 mM MgCl$_2$, 2.5 mM CaCl$_2$, and 10 mM glucose, equilibrated with 95% $O_2$ and 5% $CO_2$ (pH = 7.4, 300 mOsm). Experiments were performed at room temperature (22°C). Whole-cell current-clamp recordings were obtained with an intracellular solution containing 120 mM K-gluconate, 20 mM KCl, 10 mM HEPES, 2 mM MgCl$_2$, 2 mM Mg$_2$ATP, 0.3 mM NaGTP, 7 mM phosphocreatine, 0.6 mM EGTA, and 0.04 mM AlexaFluor594 (pH = 7.2, 295 mOsm; Thermo Fisher Scientific). Targeted cells were located at a depth of 20 to 70 μm from the surface of the slice. Neurons targeted for recordings had ovoid shapes, recognizable neuronal features such as dendrites and axons, and were not excessively bright. Glia could usually be discerned based on their more diffuse shapes and very bright somata, likely due to their more hyperpolarized membrane potential. Some glia were recorded by mistake but were easily recognizable by their very hyperpolarized membrane potential, low membrane resistance, and absence of action potential firing upon depolarization. No other criteria were used to exclude cells from characterization or analysis. Electrophysiological signals were acquired with a Multiclamp 700B amplifier (Molecular Devices). Data were shortpass filtered at 2 kHz and digitized at 10 kHz with a Digidata 1440A (Molecular Devices). Recordings were performed with the Clampex 11.0 software (Molecular Devices). Resting membrane potential, capacitance, and input resistance were measured in whole cell mode using pClamp software (Molecular Devices). Resting membrane potential was measured under the condition of no imposed current. Input resistance and capacitance were calculated by repetitively imposing a square 5 mV step in voltage-clamp configuration. Input resistance was measured in the steady-state phase of the current measurement. Membrane capacitance was measured by exponential fitting of the decay phase of the current trace in response to the voltage step. APs were evoked with 2- to 4 ms current pulses of 1.0 to 1.5 nA. EPSP and IPSP waveforms of 5, 10, 15, and 20 mV peak amplitude (*Figure 5A,D*) were applied in voltage-clamp mode. Electrophysiological data were analyzed using Clampfit 10.2 (Molecular Devices) and Igor Pro 6.32 (WaveMetrics, Portland, OR).

## Organotypic hippocampal slice cultures – random-access multi-photon (RAMP) voltage imaging

Imaging was performed with a custom-built random-access two-photon microscope (*Otsu et al., 2008*), as described before (*Chamberland et al., 2014*) and illustrated in *Figure 4A*. Briefly, the beam from a titanium:sapphire Chameleon Ultra laser (80 MHz repetition rate, pulse width of 140 fs, wavelength set at 900 nm, Coherent) was directed to a pair of acousto-optic deflectors (AODs, A-A Opto Electronics, France). This configuration permits the ultrafast redirection of the laser beam in two dimensions over the whole field of view. Due to equipment limitation, we imaged at 900 nm, a wavelength at which the two-photon brightness of ASAP1 and ASAP2s is ~18% lower than at 920 nm, the peak excitation wavelength (*Figure 1E*). The laser beam was attenuated to the desired power using an acousto-optic modulator, and then focused on the sample using a 25 × 0.95 NA water-immersion objective mounted on an upright microscope (Leica). Transmitted photons were collected and shortpass filtered at 720 nm. Emitted photons were separated into two channels by a 580 nm dichroic mirror (Semrock, Rochester, NY) and passed through a 530/60 nm bandpass filter for the green channel (ASAP1, ASAP2s, ASAP2f or Ace2N-4AA-mNeon) and 730/70 nm bandpass filter for the red channel (AlexaFluor594). Non-descanned photon counting was done with two GaAsP photomultiplier tubes (H7422P-40, Hamamatsu). Optical data were acquired with homemade software written in LabVIEW (National Instruments, Austin, TX). Dwell time was set to 50 µs per pixel in all experiments. The time to move the laser beam from one point to another in the field of view was approximately 6.5 µs, regardless of the distance between the points. Therefore, the number of points recorded dictated the scanning speed. For each cell, recorded voxels were manually distributed in an even pattern across the somatic plasma membrane. All experiments were performed at room temperature (22°C).

## Organotypic hippocampal slice cultures – data analysis

Images shown were globally adjusted for brightness and contrast, but individual portions of images were not modified in any way. Optical signals were exported using a custom-made routine in Lab-VIEW (National Instruments) and analyzed in Igor Pro (WaveMetrics). Distance measurements of the recording sites from the cell body were performed using ImageJ 1.47 (*Schneider et al., 2012*). Traces longer than 1 s were corrected for photobleaching. The decay phase of optical signals was fitted in Igor Pro with a monoexponential function. The fractional fluorescence change ($\Delta F/F$) corresponds to $(F – F_0)/ F_0$, where $F_0$ corresponds to the baseline (resting) fluorescence. The SNR was measured by dividing the peak $\Delta F/F$ by the standard deviation of the noise. For these measurements, $F_0$ was measured as the average fluorescence over the 110 to 10 ms preceding the action potential trigger. The standard deviation of the noise was measured 200 ms before the stimulus onset.

### Detectability (d′) analysis

All d′ analyses were performed on traces acquired at 925 Hz, corresponding to 20 voxels per cell and 50 µs dwell time per pixel. These traces were generated by imaging the response to a single spike followed by a 20 Hz train of 5 action potentials, or to a 10, 30, 100, and 200 Hz spike train without an isolated spike. Fluorescence was acquired by a photomultiplier tube in photon-counting mode. Single-cell optical traces were obtained by summing all 20 voxels from each cell. Single-voxel traces used the brightest of the 20 voxels recorded for each neuron.

To calculate spike detectability (d′), we first corrected traces for photobleaching. Sections of optical traces corresponding to resting fluorescence (when spikes were not occurring) were identified and fitted with a single exponential using the exp2fit function in MATLAB (Mathworks), which creates the fit using nonlinear least squares. The resulting exponential curve was used to generate a photobleaching correction function. First, the value in the exponential curve corresponding to the start of the first spike was subtracted from all points in the curve. The resulting vector was then multiplied by −1 and added to the actual fluorescence trace. This procedure corrects for photobleaching in a way that keeps the average photon count of the resting state constant across the trace and equivalent to the value immediately before the start of the first spike. This was necessary because the discriminability index assumes optical traces with constant baselines and require an actual

photon count rather than a relative fluorescence intensity (*Wilt et al., 2013*). AP detectability (d') was calculated using the formula

$$d' = \left(\frac{\Delta F}{F}\right)\sqrt{\frac{F_0\tau}{2}}$$

described by Wilt and colleagues (*Wilt et al., 2013*). To measure the fractional fluorescence response (ΔF/F) to single, isolated spikes, the difference between the peak of the action potential and the resting fluorescence ($F_0$, measured as above) was taken and divided by $F_0$. Tau (τ) was calculated by fitting the trace from the peak to baseline as described above.

Since the framework in Wilt and colleagues (*Wilt et al., 2013*) was designed for single spikes, we extended its use for spike trains by recalculating tau and $F_0$ for each for each spike in the spike train. For spike trains of 10, 20 and 30 Hz, we calculated $F_0$ by averaging the maximal fluorescence between each spike; all spikes in a train therefore shared the same $F_0$. For 100-Hz spike trains, the baseline fluorescence changed more significantly during the course of a train, so $F_0$ was calculated for each spike individually as the maximal fluorescence over the following inter-spike interval. 100-Hz spike trains sometimes produced peaks that could not be accurately located or fitted. As these peaks were eliminated from analysis, the calculated d' overestimates detectability over the entire sample. 72% of spikes from 100-Hz trains were included for ASAP1 (18 out of 25), and 60% were included for ASAP2s. The decay phase of optical signals was estimated in MATLAB (Mathworks) as described above.

False-positive and true-positive rates were calculated as follows. First, we defined a Gaussian noise distribution with a mean of 1 and a standard deviation of 1. We also defined a Gaussian signal distribution with a mean of 1 + d' and a standard deviation of 1. A simulated vector of data points was generated by randomly sampling from these distributions. This vector was 1 million samples long, and every tenth sample was drawn from the signal distribution, while all other samples were drawn from the noise distribution. The distributions were reset after being drawn upon. For this simulated data set, spikes were assumed to be composed of only a single data point. Equation (6) in *Wilt et al. (2013*) was used to build the spike log-likelihood distribution. The number of samples (N) set to 1, since our spikes are one sample long. Each point in the log likelihood distribution (L(i)) was therefore computed as follows:

$$L(i) = f(i) * \log(Sn/B) - Sn + B$$

where f(i) is the value of the simulated data set, Sn is the mean of the signal distribution, and B is the mean of the noise distribution. L(i) was computed for 'i' up to 1 million. Each point in L corresponds to the log likelihood of a spike having occurred at its corresponding point in the simulated data. An iterative algorithm was then implemented to increment a threshold in this log likelihood vector such that values above this threshold indicate the presence of a spike, and values below indicate that a spike has not occurred. False-positive and true-positive rates calculated for each threshold. The ideal threshold was chosen to be the one that maximized 'true positives - false positives' across all positions of the simulated data.

## Statistical analyses

Results presented in the form x ± y represent the mean ± SEM unless indicated otherwise. Sample sizes reported in the text are individual cells (biological replicates) unless indicated otherwise. Statistical comparisons of pre-identified measures of interest between two data sets were performed with the Student's t-test unless otherwise indicated. Prior to performing such statistical comparisons, the Shapiro-Wilk method was used to test the null hypothesis that the data followed a Gaussian (normal) distribution. When this normality hypothesis could not be rejected, Student's t-tests were performed; otherwise, the Mann-Whitney U nonparametric test was used. Prior to performing t-tests, we also tested the null hypothesis of equal variance between the two data sets, and employed Welch's correction when the null hypothesis was rejected. Statistical tests of normality and equal variance were performed with a significance level (α) of 0.05. In figure panels, p-values are graphically depicted as: *p<0.05, **p<0.01, ***p<0.001. When analyzing the results of a specific performance test (e.g. max fluorescence response to APs), we applied the Bonferroni or Holm-Bonferroni correction to the significance levels if more than one pairwise comparison was calculated, as indicated in

the corresponding figure legends. Statistical tests were performed in Excel (Microsoft) and MATLAB (MathWorks).

## Acknowledgements

We thank Y Geng, M Hintze, and S Ganesan (Stanford), and Hamdan Hamdan (Baylor College of Medicine) for providing dissociated neurons, and G-R Li for generously donating their HEK293-Kir2.1 cell line. We thank Paul De Koninck (CERVO Brain Research Centre – Université Laval) for generously providing organotypic slices and for developing the bulk gene transfer technique. We thank A Olsen of the Stanford Neuroscience Microscopy Service (supported by NIH NS069375) and TJ Vadakkan of the Optical Imaging and Vital Microscopy Core (Baylor College of Medicine) for training on two-photon microscopy systems. Cytometry data were collected on an instrument in the Stanford Shared FACS Facility supported by NIH shared instrument grant S10RR027431-01. Finally, we thank members of the St-Pierre lab for critical reading of this manuscript.

## Additional information

### Competing interests

MZL, FS-P: Holds a US patent for a voltage sensor design based on ASAP-family indicators (patent number US9606100 B2). The other authors declare that no competing interests exist.

### Funding

| Funder | Grant reference number | Author |
| --- | --- | --- |
| American Heart Association | 16POST31150011 | Haodi Wu |
| Burroughs Wellcome Fund | | Michael Z Lin |
| Canadian Institutes of Health Research | MOP-81142 | Katalin Toth |
| Natural Sciences and Engineering Research Council of Canada | RGPIN-2015-06266 | Katalin Toth |
| Natural Sciences and Engineering Research Council of Canada | Graduate fellowship | Simon Chamberland |
| National Institutes of Health | 1U01NS090600 | Michael Z Lin |
| National Institutes of Health | K99 HL133473 | Haodi Wu |
| National Institutes of Health | HL133272 | Joseph C Wu |
| National Institutes of Health | R01 EY022638 | Thomas R Clandinin |
| National Institutes of Health | R21 NS081507 | Thomas R Clandinin |
| National Science Foundation | 1707359 | François St-Pierre |
| Rita Allen Foundation | | Michael Z Lin |
| Robert and Janice McNair Foundation | McNair Medical Institute at the Robert and Janice McNair Foundation | François St-Pierre |
| Stanford University | Graduate and Interdisciplinary Graduate Fellowships | Helen H Yang |
| Stanford University | Stanford Neuroscience Microscopy Service pilot grant | Michael Z Lin François St-Pierre |
| The Walter V. and Idun Berry | Postdoctoral Fellowship | Ying Yang |
| Université Laval Neuroscience Thematic Research Centre | PhD Fellowship | Simon Chamberland |

The funders had no role in study design, data collection and interpretation, or the decision to submit the work for publication.

## Author contributions

SC, Conceptualization, Formal analysis, Investigation, Visualization, Writing—original draft, Project administration, Writing—review and editing; HHY, Conceptualization, Software, Formal analysis, Investigation, Visualization, Writing—original draft, Project administration, Writing—review and editing; MMP, SG, YY, CS, HW, Investigation; SWE, Software, Formal analysis, Visualization, Methodology; MC, Investigation, Visualization; JCW, Supervision, Funding acquisition; TRC, KT, Conceptualization, Supervision, Funding acquisition, Project administration, Writing—review and editing; MZL, Conceptualization, Supervision, Funding acquisition, Visualization, Writing—original draft, Project administration, Writing—review and editing; FS-P, Conceptualization, Software, Formal analysis, Supervision, Funding acquisition, Investigation, Visualization, Writing—original draft, Project administration, Writing—review and editing

## Author ORCIDs

Helen H Yang, http://orcid.org/0000-0001-5140-9664
Sihui Guan, http://orcid.org/0000-0003-4276-2671
Charleen Salesse, http://orcid.org/0000-0001-9520-2649
Thomas R Clandinin, http://orcid.org/0000-0001-6277-6849
Katalin Toth, http://orcid.org/0000-0002-2300-4536
Michael Z Lin, http://orcid.org/0000-0002-0492-1961
François St-Pierre, http://orcid.org/0000-0001-8618-4135

## Ethics

Animal experimentation: Animal experiments were performed in accordance with either (1) the recommendations in the Guide for the Care and Use of Laboratory Animals of the National Institutes of Health and the guidelines of the Stanford Institutional Animal Care and Use Committee under Protocol APLAC-23407, or (2) the guidelines for animal welfare of the Canadian Council on Animal Care and protocols approved by the Université Laval Animal Protection Committee (protocol number 2014-149-3). All surgery was performed under sodium pentobarbital anesthesia, and every effort was made to minimize suffering.

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
