## [Decision Letter]

Thank you for submitting your article "Fast two-photon imaging of subcellular voltage dynamics in neuronal tissue with genetically encoded indicators" for consideration by *eLife*. Your article has been reviewed by two peer reviewers, and the evaluation has been overseen by a Reviewing Editor and Eve Marder as the Senior Editor. The reviewers have opted to remain anonymous.

The reviewers have discussed the reviews with one another and the Reviewing Editor has drafted this decision to help you prepare a revised submission.

Summary:

This paper presents a novel GEVI, ASAP2s and its application in a variety of contexts: cultured neurons, human iPSC-derived cardiomyocytes, *Drosophila* visual system, and organotypic slice culture. The authors apply random-access point-scanning 2P microscopy to achieve high time resolution and decent SNR in the *Drosophila* and organotypic slice culture systems. The technical quality of the work and the presentation are high. Overall, this is a well-executed study, showing the utility of ASAP2s in a variety of systems. Importantly, they show modest, but real improvements to enable 2P functional imaging.

Essential revisions:

1) One main concern is that it is not at all clear what the relationship is between this report, featuring ASAP2s, and the 2016 study by a similar group of authors (Cell, 2016, 166, p245) that discloses ASAP2f. Clearly the mutations are different, but there is no comparison between 2s and 2f. Since this is largely the same group of authors, they should have access to the appropriate DNA constructs and even the required genetically-modified organisms. At the very least, some discussion of the relative benefits of 2s vs. 2f should be discussed. One does not need to be the clear "winner" – it's good to have a variety of methodologies to employ, but this paper does not clear up the confusion over which plasmid a prospective user would request. These considerations should be addressed: ideally with experiments and at the very least with a discussion. There should also be comparisons to the other "state of the art" indicator, mAce-Neon, out of the Schnitzer efforts. Since there is a shared co-author on these studies, this should be feasible.

2) The new GEVI is somewhat more sensitive and somewhat slower than ASAP1. This tradeoff leads to improved spike detection abilities under the conditions tested, though at the expense of spike waveform fidelity. What is the upper limit on spiking rate that can be detected by this new, slower ASAP2? For a long time the problem in genetically encoded voltage indicators (aside from poor membrane trafficking) was prohibitively slow kinetics. Now it appears to have shifted the other direction. The slower off rate of ASAP2s makes for brighter signals, but is it still fast enough for fast spikes? What is the upper limit (i.e. when should one turn to ASAP1 orASAP2f?) Selling the decrease in GEVI speed as a positive attribute of the new GEVI is rather misleading.

3) Does ASAP2s work better in 2P purely because of the change in time constant + improved sensitivity, or has there been a change to the photophysics of the FP itself? (i.e. is the FP brighter, either in quantum yield or 2P absorption?)

How was the membrane capacitance and other items in Figure 3—figure supplement 2 calculated? The capacitance would be the value I would expect to rise, upon insertion of voltage-sensing domains in a plasma membrane. It looks like the numerical value of ASAP2s's capacitance is higher than that of untransfected cells (spread of data on untransfected bar is larger).

4) A more nuanced discussion of the photobleaching under 2P illumination would be helpful. For a given illumination intensity, NA, pulse duration, and wavelength, one would expect the photobleaching rate to be proportional to the duty cycle of illumination at each pixel, i.e. the fraction of the time the laser resides on the pixel. For the photobleaching measurements in Figure 1—figure supplement 3, the duty cycle is ~10^-6 (0.4 us per pixel x 2.23 Hz). For the applications in slice culture, the duty cycle is ~0.04 (50 us per voxel x 925 Hz). This is a 40,000-fold difference in illumination dose per voxel. The authors should discuss the importance of illumination duty cycle in determining the photobleaching rate.

Surprisingly, the fast photobleaching time-constants only differed by a factor of ~100 (5 s in the table in Figure 1—figure supplement 3 vs. 0.04 s in Figure 3—figure supplement 2). This discrepancy deserves comment.

The presence of multiple time-scales in the photobleaching deserves some comment and analysis. One suspects the fast timescale represents photobleaching of the molecules directly under the laser focus, while the slower timescale represent depletion of surrounding molecules as they diffuse into the focus. If this guess is correct, then the authors could partially 'restore' the higher fluorescence of the cell either by briefly shutting the laser (so that GEVI concentration gradients can disperse), or by shifting the illumination spots to new locations. A demonstration that one can partially counteract photobleaching by these approaches would help allay some of the concerns about 2P voltage imaging.

It is not a foregone conclusion that 2P imaging is the best approach to voltage imaging in tissue. The number of points that can be multiplexed at high time resolution is limited, and photobleaching is a serious concern. A more detailed discussion of the photobleaching issues could help advance this discussion substantively.

5) The paper should also be more specific in the Abstract and throughout that the rodent "brain tissue" experiments are in organotypic slice cultures. Don't bury this fact. In many places the manuscript simply refers to "brain slice" or "intact brain tissue". The authors should exclusively refer to "organotypic slice culture." From the perspective of neuroscience applications, there is a world of difference between an acute slice and a cultured slice. These are also different from an optics perspective: cultured slices are thinner and sparser, offering much better imaging quality than acute slices. The impact of the work would have been greater had the authors looked in acute slice and e.g. characterized how deep they could image.

There have been many applications of GEVIs in organotypic slice culture with 1P imaging. It is not obvious that 2P imaging in organotypic slice cultures enables applications that could not have been achieved with more widely available 1P techniques.

---

## [Author Response]

*Essential revisions:*

*1) One main concern is that it is not at all clear what the relationship is between this report, featuring ASAP2s, and the 2016 study by a similar group of authors (Cell, 2016, 166, p245) that discloses ASAP2f. Clearly the mutations are different, but there is no comparison between 2s and 2f. Since this is largely the same group of authors, they should have access to the appropriate DNA constructs and even the required genetically-modified organisms. At the very least, some discussion of the relative benefits of 2s vs. 2f should be discussed. One does not need to be the clear "winner" – it's good to have a variety of methodologies to employ, but this paper does not clear up the confusion over which plasmid a prospective user would request. These considerations should be addressed: ideally with experiments and at the very least with a discussion. There should also be comparisons to the other "state of the art" indicator, mAce-Neon, out of the Schnitzer efforts. Since there is a shared co-author on these studies, this should be feasible.*

We agree with the reviewers that additional comparisons with ASAP2f and Ace2N-mNeon would be informative. For context, these sensors were not well characterized (the “fly-optimized” ASAP2f sensor) or published (the FRET-opsin indicator Ace2N-mNeon) when we started this work, hence our initial choice of other voltage indicators as reference GEVIs. Minor clarification: there are actually no shared co-authors with the Ace2N-mNeon Science paper from the Schnitzer lab.

To address the reviewers’ concerns, we performed multiple new experiments to compare ASAP2s to ASAP2f and Ace2N-mNeon. First, we evaluated the response of Ace2N-mNeon in *Drosophila* L2 axon terminals. We observed a minimal (but detectable) response (Figure 3), consistent with a previous study showing reduced sensitivity of other opsin-based indicators under two-photon illumination (Brinks D, Klein AJ, Cohen AE (2015). Biophys J 109:914–921). We also expressed this indicator in organotypic slice cultures and found that a significant fraction of the fluorescence appeared cytoplasmic (Figure 4—figure supplement 4), as we also observed in dissociated neuronal cultures (Figure 2—figure supplement 2). We next quantified its fluorescence responses to step depolarizations under random-access multiphoton microscopy. Consistent with our results in *Drosophila*, Ace2N-mNeon performed poorly, only producing a ~2% fractional fluorescence change to 100-mV step depolarizations (Figure 4—figure supplement 4).

We next compared ASAP2f with ASAP1 and ASAP2s. We first confirmed that all ASAP indicators are well localized at the plasma membrane in dissociated neuronal cultures (Figure 2—figure supplement 2). Second, given that its performance in *Drosophila* L2 axon terminals has already been reported (Yang HH, et al. (2016) Cell 166:245–257), we did not duplicate the results here but instead mentioned those results in the main text.

We next sought to comprehensively benchmark ASAP2f in our random-access multiphoton microscopy experiments with organotypic slice cultures. Under identical conditions, we observed that ASAP2f performed similarly to ASAP1 across all metrics, including brightness (Figure 4—figure supplement 3), peak response amplitude to evoked APs (Figure 4—figure supplement 3), kinetics (Figure 6—figure supplement 2), signal-to-noise ratio when reporting single APs (Figure 7—figure supplement 1), and peak response amplitude to individual spikes within AP trains (Figure 8—figure supplement 2).

*2) The new GEVI is somewhat more sensitive and somewhat slower than ASAP1. This tradeoff leads to improved spike detection abilities under the conditions tested, though at the expense of spike waveform fidelity. What is the upper limit on spiking rate that can be detected by this new, slower ASAP2? For a long time the problem in genetically encoded voltage indicators (aside from poor membrane trafficking) was prohibitively slow kinetics. Now it appears to have shifted the other direction. The slower off rate of ASAP2s makes for brighter signals, but is it still fast enough for fast spikes? What is the upper limit (i.e. when should one turn to ASAP1 orASAP2f?) Selling the decrease in GEVI speed as a positive attribute of the new GEVI is rather misleading.*

We agree with the reviewers that an analysis of spike rate detection would be helpful for users choosing between different members of the ASAP family of GEVIs for imaging faster-spiking neurons. To address this question, we first evoked action potentials in organotypic hippocampal slices at 10, 30, and 100 Hz. We then measured the optical response to each action potential (AP) in the train. To quantify the results, we expanded a previous analytical framework for spike detection (Wilt BA, Fitzgerald JE, Schnitzer MJ (2013). Biophys J 104:51–62) to analyze spike trains rather than individual spikes. Results are described in Figure 8 and Figure 8—figure supplement 1,Figure 8—figure supplement 2. As expected, detectability of spikes within trains decreased with increasing train frequency. ASAP2s performed better than ASAP1 at frequencies of 30Hz or below, but their performance was nearly matched (but low) at 100 Hz. This is largely because ASAP2s’ improved sensitivity balanced its slower kinetics at this spike train frequency.

*3) Does ASAP2s work better in 2P purely because of the change in time constant + improved sensitivity, or has there been a change to the photophysics of the FP itself? (i.e. is the FP brighter, either in quantum yield or 2P absorption?).*

For spike detection, the improvements of ASAP2s are largely due to improvements in sensitivity, and slower off-kinetics. Using the d’ metrics as our measure of detectability (Wilt BA, Fitzgerald JE, Schnitzer MJ (2013)), we quantified the contribution of the parameters leading to ASAPs’ better detectability. d’ is proportional to the sensitivity (ΔF/F), to the square root of the off-kinetics time constant, and to the square root of the brightness. Compared to ASAP1, the improvement to d’ for spike detection with ASAP2s is therefore predicted to be, on average, just under two-fold from gains in sensitivity and over two-fold due to larger off-kinetics time constants. We did not detect a statistically significant change in the molecular brightness of ASAP2s over ASAP1 in HEK293 cells (Figure 1—figure supplement 2) or of cellular brightness in organotypic slices (Figure 4).

*How was the membrane capacitance and other items in Figure 4—figure supplement 1 calculated? The capacitance would be the value I would expect to rise, upon insertion of voltage-sensing domains in a plasma membrane. It looks like the numerical value of ASAP2s's capacitance is higher than that of untransfected cells (spread of data on untransfected bar is larger).*

Resting membrane potential, capacitance and, input resistance were calculated in whole cell mode using pClamp software (Molecular Devices), with additional details provided in the Materials and methods. We agree with the reviewers that an increase in capacitance is predicted given the charges present on voltage indicators. However, this increase appears to be sufficiently small such that the capacitance of cells expressing voltage indicators was not statistically different from that of untransfected cells. This observation is consistent with other reports showing no or modest changes in variables that depend in part on membrane capacitance (as reviewed in: Yang HH, St-Pierre F (2016). J Neurosci 36:9977–9989). However, given that a previous study using the (related) voltage indicator ArcLight reported an increase in capacitance, we added a sentence to the text recommending users to re-evaluate membrane electrical properties when changing indicator expression conditions. We also improved the description of this data by converting our bar plots to box and whisker plots. We also added data for ASAP2f given the new experiments we performed with this indicator in this revised manuscript.

*4) A more nuanced discussion of the photobleaching under 2P illumination would be helpful. For a given illumination intensity, NA, pulse duration, and wavelength, one would expect the photobleaching rate to be proportional to the duty cycle of illumination at each pixel, i.e. the fraction of the time the laser resides on the pixel. For the photobleaching measurements in Figure 1—figure supplement 4, the duty cycle is ~10^-6 (0.4 us per pixel x 2.23 Hz). For the applications in slice culture, the duty cycle is ~0.04 (50 us per voxel x 925 Hz). This is a 40,000-fold difference in illumination dose per voxel. The authors should discuss the importance of illumination duty cycle in determining the photobleaching rate.*

*Surprisingly, the fast photobleaching time-constants only differed by a factor of ~100 (5 s in the table in Figure 1—figure supplement 4 vs. 0.04 s in Figure 6—figure supplement 1 ). This discrepancy deserves comment.*

We agree with the reviewers that a more comprehensive discussion and characterization of photobleaching would be helpful. We have therefore performed several new experiments (Figure 1—figure supplement 3, Figure 1—figure supplement 4), and greatly expanded our discussion of photostability in the main text. First, we tested how photobleaching kinetics varied with two-photon laser power, keeping all other imaging parameters constants (Figure 1—figure supplement 4). As suspected, the response was nonlinear. It was also very complex, with lower-power curves being better described with an additional time constant. As suggested, we also discussed the impact of pixel duty cycle on photobleaching. We further show how the pixel duty cycle and laser power should be considered together (i.e. that a key determinant is the power per pixel). Specifically, two conditions with different laser power, frame rate, and dwell time – but identical power per pixel – produced comparable photobleaching (Figure 1—figure supplement 4). We also carefully characterized reversible photobleaching, as discussed later.

We also sought to address the specific example mentioned by the reviewers. Including differences in power levels and given a correction of a pixel dwell time (Figure 1—figure supplement 4), the difference in power per pixel between the two assays is ~8,000. This is indeed larger than the difference in time constants. However, comparing these two conditions is extremely difficult for at least three reasons:

a) If two-photon photobleaching is at least proportional to the square of the laser power (e.g. Oheim et al. 2001. J Neurosci Methods 111:29–37), then the fast component of Figure 1—figure supplement 4. would be orders of magnitude smaller than the ~2 ms exposure time per image of Figure 6—figure supplement 1. As a result, it would be detected as photobleaching and would simply appear as a lower brightness in the first data point. In other words, the fast photobleaching components of the two experiments may not be functionally equivalent.

b) The difference in power per pixel is so large that GFP photobleaching mechanisms may be mechanistically different. For example, over a much smaller range of power, we already see that the number of exponential terms needed to fit photobleaching kinetics changes depending on laser power (Figure 1—figure supplement 4).

c) The two experiments differ in hardware, microscopy technique, and biological preparation. For example, imaging in Figure 1—figure supplement 4 was performed by continuously moving the laser (raster scanning), and used a 60X objective. Meanwhile Figure 6—figure supplement 1 was performed by rapidly moving the laser beam using Acousto-Optic Deflectors to only image selected cellular voxels, and used a 25X objective. These differences will affect, for instance, the actual irradiance (power per unit area). Another important difference is that laser pulses of Figure 1—figure supplement 4 were pre-compensated for dispersion of laser pulses in the microscope optical path, while those of Figure 6—figure supplement 1 were not. Moreover, scattering of the excitation light is expected to be greater for experiment Figure 6—figure supplement 1, which was performed on cells *in tissue*, compared to Figure 1—figure supplement 4, which was performed on dissociated cells in media. Finally, the extracellular environment is different between the two conditions, a parameter which may be relevant given that the GFP in ASAP is extracellular and circularly permuted and might therefore be sensitive to its environment.

Given the above, the Discussion now recommends users to re-evaluate some performance metrics when deploying indicators in new experimental preparations or when using different illumination conditions.

*The presence of multiple time-scales in the photobleaching deserves some comment and analysis. One suspects the fast timescale represents photobleaching of the molecules directly under the laser focus, while the slower timescale represent depletion of surrounding molecules as they diffuse into the focus. If this guess is correct, then the authors could partially 'restore' the higher fluorescence of the cell either by briefly shutting the laser (so that GEVI concentration gradients can disperse), or by shifting the illumination spots to new locations. A demonstration that one can partially counteract photobleaching by these approaches would help allay some of the concerns about 2P voltage imaging.*

*It is not a foregone conclusion that 2P imaging is the best approach to voltage imaging in tissue. The number of points that can be multiplexed at high time resolution is limited, and photobleaching is a serious concern. A more detailed discussion of the photobleaching issues could help advance this discussion substantively.*

Fast and slow photobleaching time constants are observed with superfolder GFP, the parental protein of the GFP in ASAP indicators (Pédelacq J-D et al. (2006) Nat Biotechnol 24:79–88), suggesting that at least some of ASAP1/ASAP2s’ multiple photobleaching time-scales are an intrinsic molecular property. The revised manuscript now references this study to provide a context for the multiple exponentials observed.

We next followed the reviewers’ recommendation to quantify fluorescence following illumination gaps. Under both one- and two-photon illumination, we observed a partial recovery in fluorescence following a period of incubation in darkness (Figure 1—figure supplement 3 and Figure 1—figure supplement 4). This fluorescence recovery may represent dark reversion of bleached to unbleached molecules, diffusion of unbleached probes to the illuminated area, or both. Illumination gaps could therefore provide a strategy for partially mitigating photobleaching in experiments that can tolerate regular incubations in the dark.

We also note that our RAMP imaging system excludes the vast majority of available photons per cell by only recording from a smaller number of small voxels. In cases where subcellular resolution is not needed, expanding the point spread function of the illumination beam could enable excitation of an entire soma rather than a small number of voxels. This approach would require lower light power to produce identical total fluorescence, thereby reducing photobleaching kinetics. We incorporated this possible solution in the Discussion, and mentioned that further improvement in indicator photostability would be critical for long-term imaging.

*5) The paper should also be more specific in the Abstract and throughout that the rodent "brain tissue" experiments are in organotypic slice cultures. Don't bury this fact. In many places the manuscript simply refers to "brain slice" or "intact brain tissue". The authors should exclusively refer to "organotypic slice culture."*

We agree that “organotypic slice culture” is more specific and have made the recommended changes in the text. We have kept more generic terms (e.g. “brain tissue”) when discussing the use of voltage indicators more generally, encompassing our results from both in vivo (fly) and organotypic slice culture preparations.

*From the perspective of neuroscience applications, there is a world of difference between an acute slice and a cultured slice. These are also different from an optics perspective: cultured slices are thinner and sparser, offering much better imaging quality than acute slices. The impact of the work would have been greater had the authors looked in acute slice and e.g. characterized how deep they could image.*

We agree with the reviewers that acute slices are a very valuable experimental preparation, and we hope to confirm that our results extend to acute slices in a future manuscript. For context, we imaged neurons at depths between 20 and 70 µm from the surface, avoiding the first 20 µm as done when recording from acute slices to avoid lower quality neurons (Booker et al. (2014) J Vis Exp 91:51706). This recording depth range is also comparable to what is commonly done when patch-clamping cells in acute slices (Okada Y. (2012) Patch clamp techniques: from beginning to advanced protocols). Patching cells at depths exceeding 100 µm is challenging because cells and pipette electrodes become difficult to visualize using transmitted light at deeper locations. While two-photon imaging enables excitation of fluorophores at greater depths, this manuscript sought to compare our optical signals with the ‘ground truth’ provided by simultaneous electrical recordings.

Of note, since our detector is located below the slice rather than above the objective (Figure 4). Given our slice thickness of ~150 µm, and imaged neurons being 20-70 µm from the top surface of the slice, emitted photons will have to travel through tissue with thickness 80-130 µm thick prior to reaching the detector. Many upright imaging systems have their detectors located above the objective, so that SNR will vary primarily based on the depth of the imaged cell and not the slice thickness.

Although obviously different from acute slices, organotypic slice cultures are considered valuable preparations by many scientists to study the hippocampus because they retain important physiological attributes. For example, hippocampal principal neurons in organotypic slices maintain a normal cytoarchitecture, grow in layers (Stoppini L et al. (1991) J.Neurosci Methods 37:173-182) and present comparable dendritic morphology to age-matched acute slices (De Simoni A, Griesinger CB, Edwards FA (2003). J Physiol 550:135–147). In addition, compared to acute slices, organotypic slices present the advantage of being more easily amenable to transfection techniques. For these reasons, we choose to perform our experiments in organotypic slice cultures.

Overall, we agree with the reviewers that acute slices are a valuable preparation that we should evaluate in future work, given that they better preserve endogenous neuronal connectivity and often present a more challenging imaging environment. Our current results demonstrating fast two-photon voltage imaging combined with patch-clamp recordings in organotypic slice cultures motivate further evaluation of voltage imaging in acute slices and in vivo.

*There have been many applications of GEVIs in organotypic slice culture with 1P imaging. It is not obvious that 2P imaging in organotypic slice cultures enables applications that could not have been achieved with more widely available 1P techniques.*

We agree that 1P voltage imaging in slices can also be valuable. However, 1P experiments with widefield imaging will typically rely on sparse expression of the sensor to avoid convolving signals from neurons located above or below the plane of focus. Signal overlap is especially problematic for indicators that are bright at the resting membrane potential, a category that includes many of the best-performing GEVIs. 2P mitigates this issue by providing greatly improved optical sectioning. 2P also enables more focal excitation, reducing phototoxicity in neurons outside of the focal area. This is an important advantage given that voltage imaging typically requires continuous illumination at relatively high power. More generally, the development and characterization of indicators under 2P imaging is critical to eventually enable deep-tissue voltage imaging in rodents.